# 3ViewSense: Spatial and Mental Perspective Reasoning from Orthographic Views in Vision-Language Models

**Shaoxiong Zhan** [* 1]  **Yanlin Lai** [* 1]  **Zheng Liu** [* 1]  **Hai Lin** [1]
**Shen Li** [2]  **Xiaodong Cai** [1]  **Zijian Lin** [1]  **Wen Huang** [1]  **Hai-Tao Zheng** [1]

## Abstract

Current Large Language Models have achieved Olympiad-level logic, yet Vision-Language Models paradoxically falter on elementary spatial tasks like block counting. This capability mismatch reveals a critical "spatial intelligence gap," where models fail to construct coherent 3D mental representations from 2D observations. We uncover this gap via diagnostic analyses showing the bottleneck is a missing view-consistent spatial interface rather than insufficient visual features or weak reasoning. To bridge this, we introduce **3ViewSense**, a framework that grounds spatial reasoning in Orthographic Views. Drawing on engineering cognition, we propose a "Simulate-and-Reason" mechanism that decomposes complex scenes into canonical orthographic projections to resolve geometric ambiguities. By aligning egocentric perceptions with these allocentric references, our method facilitates explicit mental rotation and reconstruction. Empirical results on spatial reasoning benchmarks demonstrate that our method significantly outperforms existing baselines, with consistent gains on occlusion-heavy counting and view-consistent spatial reasoning. The framework also improves the stability and consistency of spatial descriptions, offering a scalable path toward stronger spatial intelligence in multimodal systems. [1]

## 1. Introduction

The advent of Large Vision-Language Models (VLMs) has revolutionized multimodal understanding. However, a startling paradox remains: while state-of-the-art models (e.g., GPT-4o, GPT-5 class) exhibit Olympiad-level symbolic logic (Guo et al., 2025; Jaech et al., 2024; Zhan et al., 2026), they often falter on elementary spatial tasks, such as counting stacked blocks under occlusion (Cai et al., 2025). This capability mismatch reveals a critical spatial intelligence gap: models possess powerful deductive engines but lack a coherent 3D mental representation mechanism to ground their reasoning in the physical world, leading to severe performance degradation when reasoning over uncertain, partially observed spatial regions.

To identify the root cause of this gap, we conducted a diagnostic investigation. First, we question whether the visual encoder is the bottleneck. In our visual information sufficiency test, we freeze the visual features and train a lightweight probe on the block counting task. The probe achieves high accuracy ($55.8\%$) where the full VLM fails, proving that the encoder successfully extracts sufficient geometric information (detailed in Appendix C.3).

Second, we study the reasoning interface. Prior work shows that strengthening the language model (or language-only supervision) can substantially improve VLM reasoning (He et al., 2025). Accordingly, we augment the image input with an *additional* orthographic three-view context (front/left/top) generated from image descriptions. As illustrated in Figure 1, conditioning the same model on both the image and this three-view contextual description leads to a dramatic improvement in reasoning accuracy (e.g., Gemini-3-pro improves by over $30\%$ absolute, detailed in Appendix C.4). This implies that the reasoning engine is intrinsically capable but lacks a structured spatial interface to reliably access and organize the relevant visual information.

Synthesizing these findings, we hypothesize that the spatial intelligence gap stems neither from "blind" encoders nor "dumb" reasoners, but from a misalignment in the inference process: current models lack a stable *view-consistent intermediate representation* to bridge egocentric perception and logical reasoning. Without this bridge, visual features are not effectively translated into spatial concepts, leading to reasoning drift and hallucinations.

---

[1]Shenzhen International Graduate School, Tsinghua University, Shenzhen, China [2]School of Software Engineering, Chongqing University, Chongqing, China. Correspondence to: Shaoxiong Zhan <zhanshaoxiongthu@gmail.com>, Hai-Tao Zheng <zheng.haitao@sz.tsinghua.edu.cn>.

*Proceedings of the 43rd International Conference on Machine Learning*, Seoul, South Korea. PMLR 306, 2026. Copyright 2026 by the author(s).

[1]https://github.com/Jasaxion/3ViewSense

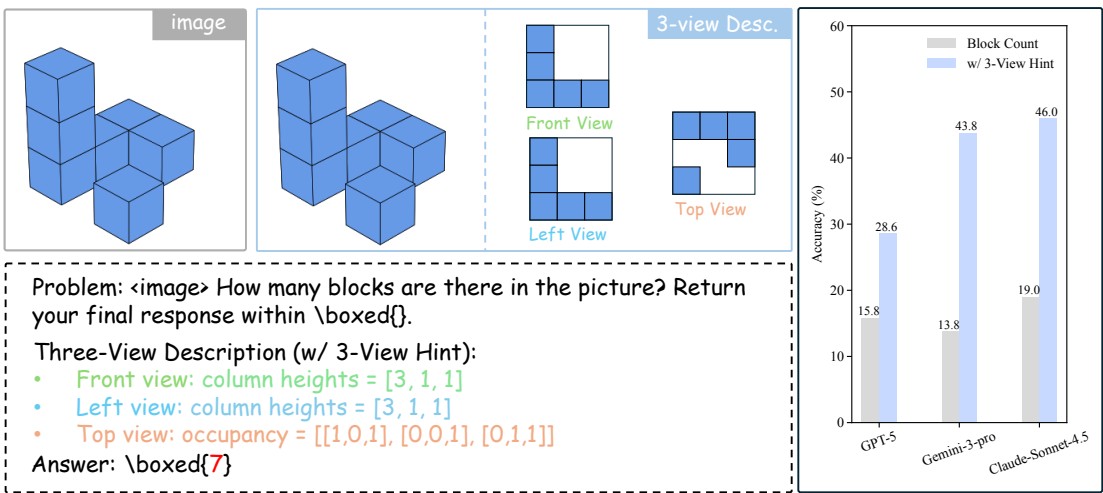

*Figure 1.* Motivation for explicit three-view reasoning. Providing explicit orthographic three-view descriptions (front/left/top) improves block-counting performance under occlusion, highlighting the role of view-consistent spatial representations.

To bridge this gap, we introduce **3ViewSense**, a framework that grounds spatial reasoning in orthographic views. Inspired by the way engineering drawings define 3D structure through standard projections, 3ViewSense follows a simulate-and-reason pipeline that first induces a view-consistent spatial representation and then performs explicit reasoning on top of it.

Concretely, 3ViewSense separates the learning objective into two stages (Section 3.3). In Stage I, Orthographic Mental Simulation (OMS) is trained to generate structured orthographic descriptions from an egocentric image. In Stage II, View-Grounded Reasoning (VGR) is trained to solve spatial queries by conditioning on the induced orthographic views and producing the final answer. Starting from the Stage II model, we further apply GRPO-based reinforcement learning to refine correctness under math-verifiable rewards while preserving view-grounded reasoning behavior. This design is motivated by our diagnostic findings: spatial reasoning becomes substantially more reliable when the model can mentally infer and complete information from other orthographic views to form a view-consistent representation.

Following the 3ViewSense framework, we train models on our in-domain dataset OrthoMind-3D to acquire orthographic-view mental simulation and view-grounded reasoning. Experiments show that 3ViewSense consistently improves reasoning accuracy on both in-domain and out-of-domain splits, and the gains also transfer to other spatial reasoning benchmarks (e.g., MindCube-Tiny: 27.2→38.9).

Our contributions are as follows: (1) We introduce **OrthoMind-3D**, a diagnostic benchmark that exposes key failure modes of spatial reasoning under occlusion and perspective shifts. (2) Based on this diagnosis, we propose 3ViewSense, a Simulate-and-Reason framework that grounds reasoning in mentally induced orthographic views. (3) 3ViewSense delivers strong accuracy gains on OrthoMind-3D (in-domain and out-of-domain) and transfers to other spatial reasoning benchmarks.

## 2. Related Work

### 2.1. Spatial Reasoning with VLMs

The landscape of spatial capabilities in Vision-Language Models (VLMs) has evolved significantly, progressing from elementary visual perception tasks (Li et al., 2025a; Bai et al., 2023) to intricate spatial reasoning that demands deep mental simulation (Yin et al., 2025; Chen et al., 2026; Lee et al., 2025). Recent efforts to bridge the spatial intelligence gap in VLMs generally fall into three paradigms.

**Auxiliary Modalities and Tool Usage.** To overcome the limitations of RGB-only inputs, works augment VLMs with 3D encoders (Wu et al., 2026a; Chen et al., 2026; Wang et al., 2025a) or fine-tune with vision-centric data like segmentation masks (Chen et al., 2025; Liu et al., 2025; Wang et al., 2025b; Fan et al., 2026; Ma et al., 2024). Beyond internal integration, some work (Zhou et al., 2025; Su et al., 2025; Wu et al., 2026b) adopts a tool-centric approach to exploit the planning and programming abilities of LLMs to actively call external vision modules as executable tools. While effective, these methods often incur high computational overhead and dependency on external models.

**Advanced Training Strategies** To elicit stronger reasoning capabilities without relying on external tools, recent studies have turned to specialized training paradigms. SpatialLadder (Li et al., 2025c) employs progressive curriculum learning, while recent Reinforcement Learning (RL) methods (Liao et al., 2025; Ouyang et al., 2025; Xu et al., 2026)

incentivize models to self-correct reasoning paths.

**Mental Modeling and Perspective Taking.** A parallel stream of research focuses on internalizing spatial understanding through Spatial Mental Models. MindCube (Yin et al., 2025) and APC (Lee et al., 2025) utilize cognitive maps or mental imagery to hallucinate plausible 3D structures from 2D inputs to overcome egocentric bias.

Distinct from methods relying on auxiliary data, unstructured RL, or implicit imagery, 3ViewSense introduces a structured "Simulate-and-Reason" mechanism grounded in orthographic views, offering a computationally efficient and geometrically rigorous path to spatial intelligence.

### 2.2. Benchmarking Spatial Reasoning

The evaluation of spatial intelligence in Vision-Language Models (VLMs) has evolved from static, object-centric recognition to dynamic, space-centric reasoning. Foundationally, CV-Bench (Tong et al., 2024) establishes vision-centric grounding by reformulating traditional vision tasks into VQA formats, while OmniSpatial (Jia et al., 2026) formalizes the transition from "Object-level" perception to high-level "Space-level" reasoning. To address multi-dimensional complexities, SPBench (Li et al., 2025c) provides a hierarchical suite spanning single-image and multi-view modalities, and ViewSpatial-Bench (Li et al., 2025b) specifically probes the gap between egocentric (camera-centered) and allocentric (entity-centered) spatial frames. Furthermore, cognitive-oriented benchmarks such as MindCube (Yin et al., 2025), Sphere (Zhang et al., 2025a), and Open3D-VQA (Zhang et al., 2025b) evaluate internal "mental models" by testing reasoning under occlusion or within unconstrained 3D environments. Despite these advances, existing benchmarks lack the diagnostic granularity needed for rigorous 2D–3D alignment, particularly for evaluating mental rotation and orthogonal projection. OrthoMind-3D is designed to address this gap.

## 3. Methodology

### 3.1. Problem Formulation

We consider the problem of spatial reasoning in Vision-Language Models (VLMs). Formally, given an egocentric 2D image $I_{ego} \in \mathcal{I}$ and a natural language query $q \in \mathcal{Q}$, the goal is to predict the correct answer $a \in \mathcal{A}$ that typically involves understanding 3D spatial relationships, object counting, or perspective taking.

**Standard VLM Inference.** Conventional VLMs approach this task by directly modeling the conditional probability distribution $P(a|I_{ego}, q)$. The optimal answer $a^*$ is obtained by maximizing this likelihood:

$$a^* = \arg\max_{a \in \mathcal{A}} P(a \mid I_{ego}, q). \tag{1}$$

However, this end-to-end formulation treats spatial reasoning as a black-box mapping. As discussed in Section 1, this is inherently ill-posed because a single 2D image $I_{ego}$ creates ambiguity regarding the underlying 3D structure (e.g., depth occlusion), often leading to spatial hallucinations when complex reasoning is required.

**3ViewSense Formulation.** To bridge this spatial intelligence gap, we propose to explicitly model the mental imagery of the scene's 3D structure. Drawing inspiration from engineering cognition, we introduce a set of latent variables $\mathcal{V} = \{v_{front}, v_{left}, v_{top}\}$, representing the canonical Orthographic Views (front, top, and left projections) of the scene.

We reformulate the reasoning process as a two-stage probabilistic framework: (1) *Mental Simulation*, where the model infers the orthographic views from the egocentric input, and (2) *View-Grounded Reasoning*, where the answer is derived based on these explicit spatial priors.

Mathematically, we decompose the objective using the chain rule of probability. The inference of answer $a$ is conditioned on both the input and the generated orthographic mental images:

$$P(a \mid I_{ego}, q) = \sum_{\mathcal{V}} P(a \mid \mathcal{V}, I_{ego}, q) \cdot P(\mathcal{V} \mid I_{ego}, q). \tag{2}$$

Since integrating over all possible view combinations is intractable, we approximate this by maximizing the joint probability through a deterministic "Simulate-and-Reason" pipeline. We first generate the most probable set of orthographic views $\hat{\mathcal{V}}$:

$$\hat{\mathcal{V}} = \arg\max_{\mathcal{V}} P_{\theta_{sim}}(\mathcal{V} \mid I_{ego}, q), \tag{3}$$

where $P_{\theta_{sim}}$ represents our Orthographic Mental Simulator. Subsequently, the final answer is predicted by reasoning over these structured views:

$$a^* = \arg\max_{a \in \mathcal{A}} P_{\theta_{reason}}(a \mid \hat{\mathcal{V}}, I_{ego}, q). \tag{4}$$

By introducing $\hat{\mathcal{V}}$, we transform the abstract 3D spatial reasoning task into a tractable pattern recognition problem on structured 2D planes, reducing geometric ambiguity.

### 3.2. Datasets Construction

To systematically develop and evaluate the spatial intelligence of Vision-Language Models, specifically their ability to perform mental perspective reasoning, we curate a diagnostic dataset named **OrthoMind-3D**. Rather than aiming

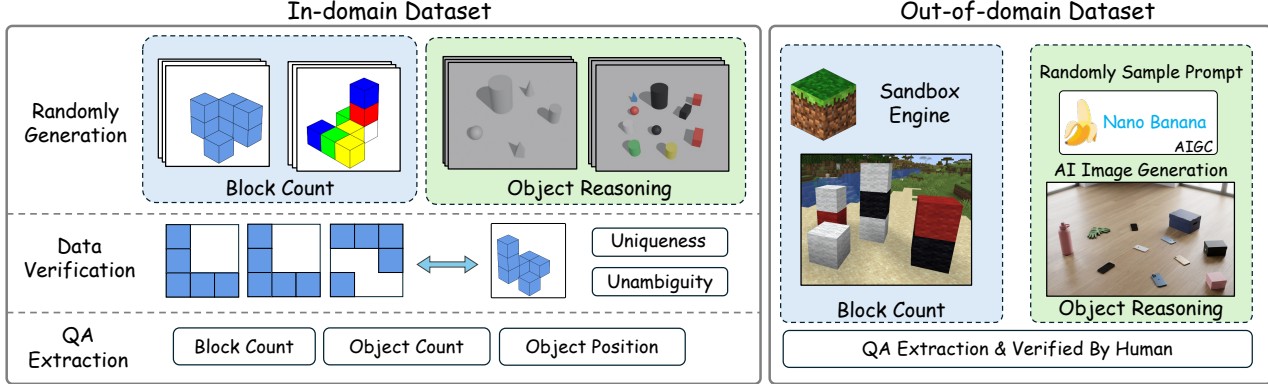

*Figure 2.* The construction pipeline of our **OrthoMind-3D** dataset. To bridge the gap between visual perception and mental spatial reasoning, we curate data from two distinct domains. For In-Domain data, we utilize programmatic synthesis with strict geometric constraints to train the model's orthographic projection capabilities. For Out-of-Domain data, we employ sandbox game engines and generative AI techniques to evaluate the model's robustness and generalization in unstructured environments.

for explicit 3D reconstruction from 2D inputs, this benchmark serves a dual purpose: (1) to rigorously evaluate the extent to which orthographic mental simulation can enhance reasoning precision and mitigate spatial hallucinations in complex geometric scenarios; and (2) to enable the model to learn this "Simulate-and-Reason" process by inferring latent orthographic views from single-view egocentric inputs to support more robust decision-making.

As illustrated in Figure 2, the data curation pipeline is bifurcated into two streams: In-Domain data synthesized for explicit orthographic training, and Out-of-Domain (OOD) data designed to assess generalization robustness. The dataset covers two primary tasks: *Block Counting*, which targets volumetric reasoning and complex occlusion handling in 3D structures, and *Object Reasoning*, which evaluates capabilities in both relative spatial positioning and general object enumeration. To ensure fine-grained analysis, both tasks are further stratified into attribute-specific (e.g., querying color/size) and single-attribute sub-tasks. For comprehensive statistics and detailed visualization examples of the collected data, please refer to Appendix B.1.

**Block Counting Task.** The core challenge in counting stacked blocks from a single viewpoint lies in the ambiguity of depth. To train a model that can reliably deduce 3D structures via orthographic views, it is imperative that the mapping from the provided three views (top, front, left) to the total cube count is bijective. For the **In-Domain** subset, we employ programmatic synthesis to generate block configurations. However, random stacking often yields ambiguous structures where multiple 3D configurations correspond to the same projections. To enforce strict bijectivity between the 3D configuration and its 2D projections, we derive a necessary and sufficient uniqueness condition. Formally, a stack configuration $H$ (where $H_{x,y}$ denotes the height at

position $x, y$) is uniquely determined if and only if every occupied position satisfies:

$$
\begin{aligned}
\forall(x,y), \quad & \underbrace{[H_{x,y} = 1 \wedge (M^c = 1 \vee M^r = 1)]}_{\text{Case I: Base-Level Dominance}} \\
\vee \quad & \underbrace{[H_{x,y} > 1 \wedge (H_{x,y} > O^c \vee H_{x,y} > O^r)]}_{\text{Case II: Multi-Level Occlusion}}
\end{aligned} \tag{5}
$$

where $M^c, M^r$ denote the global maximums of the corresponding column/row, and $O^c, O^r$ represent the maximum heights of other blocks in the same line (excluding $(x, y)$). We rigorously filter synthetic data to ensure this condition holds (proof in Appendix A.1).

For the Out-of-Domain subset, we assess robustness using a voxel-based sandbox engine. Deviating from rigid in-domain grids, blocks are stochastically scattered to form unstructured, high-entropy piles. We manually sample these scenes from diverse viewpoints to introduce natural perspective variations. Each query-answer pair is manually verified by human annotators for correctness.

**Object Reasoning Task.** This task assesses capabilities in both relative spatial positioning and object enumeration. For In-Domain data, we utilize a 3D rendering engine. Objects are arranged on a single horizontal plane to decouple spatial reasoning from vertical occlusion. We define two sub-tasks: *Object Counting* and *Object Positioning*. For positioning, we discretize spatial relations into 8 directions (e.g., "front", "front-left"). To handle ambiguity near the boundary of the axis, we treat both the cardinal and intermediate directions as valid labels if the object aligns within the $5°$ margin of a canonical axis.

For Out-of-Domain data, we synthesize photorealistic scenes using Gemini-3-Pro-Image (Google DeepMind, 2025) (Nano Banana). We employ diverse prompts spec-

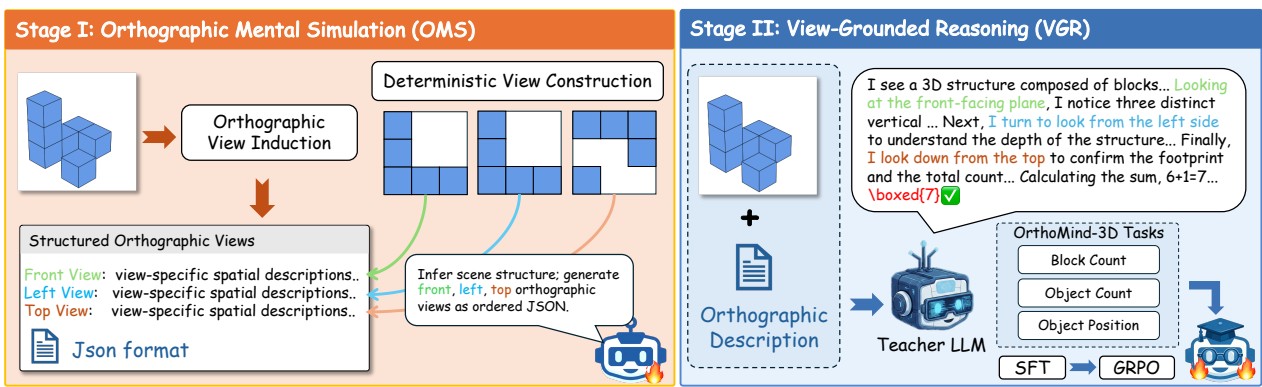

*Figure 3.* The training framework of **3ViewSense**. Stage I learns to induce canonical front, left, and top orthographic views from an egocentric input. Stage II performs view-grounded reasoning by integrating the inferred views to generate reasoning traces and final answers, while reinforcement learning further internalizes this view-based reasoning capability.

ifying random object attributes and environments. As detailed in Appendix B.2, we enforce constraints like "non-overlapping" and "high-angle perspective" to ensure task feasibility. Finally, all synthesized data undergoes manual verification to guarantee the accuracy of spatial relation labels.

### 3.3. 3ViewSense Training Framework

We propose a modular training framework that decouples *what* spatial abilities the model should acquire from *how* these abilities are optimized. Conceptually, 3ViewSense consists of two capability-oriented stages: (i) **Orthographic Mental Simulation** (OMS), which equips the model with the ability to internally infer canonical orthographic views from an egocentric observation; and (ii) **View-Grounded Reasoning** (VGR), which trains the model to leverage these inferred views to solve spatial reasoning tasks.

Figure 3 illustrates the overall training pipeline that instantiates the proposed "Simulate-and-Reason paradigm". The framework introduces an explicit intermediate representation in the form of structured orthographic views, enabling the model to first internalize spatial structure and then reason over it in a view-grounded manner.

**Stage I: Orthographic Mental Simulation (OMS).** Stage I focuses on learning the mental simulation process defined in Eq. 3, namely inducing a set of canonical orthographic views $\mathcal{V} = \{v_{\text{front}}, v_{\text{left}}, v_{\text{top}}\}$ from a single egocentric observation. In our implementation, OMS is trained via supervised fine-tuning (SFT) using programmatically extracted orthographic annotations from the synthetic In-Domain data (Section 3.2). Each view is represented as a structured description that captures view-specific spatial information. For block counting tasks, views encode visible block primitives with stacking and occlusion cues; for object reasoning tasks,

views are represented as ordered perceptual sequences (e.g., left-to-right or back-to-front scan order). An illustrative example of the three-view description and the Stage-I (OMS) instruction is provided in Appendix Figure 7. The model is optimized with standard maximum-likelihood sequence learning to generate the structured three-view representation conditioned on $(I_{\text{ego}}, q)$, yielding the Stage-I SFT model $M_{\text{stage1}}^{\text{SFT}}$.

**Stage II: View-Grounded Reasoning (VGR).** Stage II optimizes the view-grounded objective in Eq. 4, learning to predict answers by explicitly conditioning on the inferred orthographic views $\hat{\mathcal{V}}$. While $\hat{\mathcal{V}}$ provides strong structural priors, solving challenging tasks (e.g., counting under occlusion or perspective taking) further requires integrating evidence across multiple projections into a coherent 3D mental model.

**Stage-II SFT initialization.**

We first perform supervised fine-tuning to teach the model to generate three-view-grounded natural-language reasoning and the final answer. Specifically, we construct *view-grounded reasoning traces* that follow a human-like integration order (front → left → top), written in a first-person mental simulation style and avoiding explicit format references (Appendix 9). These traces are generated by a strong teacher model and filtered by the correctness of the final-answer, resulting in an initialized reasoner in SFT $M_{\text{stage2}}^{\text{SFT}}$.

**GRPO-based RL refinement.** Starting from $M_{\text{stage2}}^{\text{SFT}}$, we further refine the reasoner with Group Relative Policy Optimization (GRPO) (Shao et al., 2024). Motivated by recent findings that online RL is less prone to catastrophic forgetting than SFT and can better preserve previously learned capabilities (Shenfeld et al., 2026), we adopt GRPO to (i) mitigate potential generalization degradation under large-scale

*Table 1.* Main results on OrthoMind-3D. We report accuracy (%) for block counting and object reasoning, covering both cardinality queries and attribute-conditioned (attr.) queries, as defined in Section 4.2. "–" denotes entries where the model cannot produce outputs in the required format for evaluation.

| Model | Block Counting | | Object Reasoning | | | |
|---|---|---|---|---|---|---|
| | Block Count | Block Count (Attr.) | Object Count | Object Position | Object Count (Attr.) | Object Position (Attr.) |
| *Proprietary Models* | | | | | | |
| GPT-4o | 15.8 | 53.2 | 68.3 | 39.3 | 71.2 | 47.2 |
| Gemini-2.0-Flash | 18.2 | 54.0 | 69.7 | 62.7 | 86.4 | 62.6 |
| GPT-5 | 15.8 | 50.7 | 64.0 | 60.3 | 77.6 | 66.0 |
| Gemini-3-pro | 13.8 | **80.2** | **83.3** | **71.6** | **93.2** | **93.6** |
| Claude-Sonnet-4.5 | **19.0** | 63.4 | 26.7 | 54.3 | 61.8 | 73.4 |
| *Specialized Spatial Reasoning Models* | | | | | | |
| SpatialLadder-3B | 8.4 | 27.4 | 39.6 | 25.3 | 49.2 | 25.6 |
| Spatial-MLLM-4B (Wu et al., 2026a) | 1.8 | 24.8 | 37.7 | – | 24.4 | – |
| SpaceOm-4B (Yin et al., 2025) | 10.4 | 47.2 | 63.6 | 17.6 | 60.2 | 25.4 |
| SpaceQwen2.5-VL-3B-Instruct (Jia et al., 2026) | 10.6 | 48.6 | 55.0 | 16.0 | 63.0 | 13.4 |
| *Open Source Models* | | | | | | |
| XiaoMiMo-VL-7B-RL | 18.2 | 59.2 | 64.3 | 44.3 | 77.4 | 59.2 |
| GLM4.1V-9B | 15.0 | 42.5 | 46.6 | 39.0 | 68.6 | 46.2 |
| InternVL3.5-4B | 10.6 | 53.2 | 51.6 | 40.0 | 68.2 | 50.8 |
| InternVL3.5-8B | 15.6 | 54.5 | 55.3 | 41.0 | 73.2 | 55.8 |
| Qwen2.5-VL-7B | 9.2 | 42.1 | 63.3 | 23.6 | 70.0 | 36.4 |
| Qwen2.5-VL-3B | 10.4 | 47.4 | 58.3 | 16.6 | 60.0 | 16.6 |
| Qwen3-VL-8B-Instruct | 10.6 | 43.8 | 62.0 | 47.6 | 76.6 | 60.2 |
| Qwen3-VL-4B-Instruct | 6.2 | 43.4 | 59.0 | 41.0 | 74.8 | 45.6 |
| 3ViewSense-4B-sft (ours) | 33.4 | 63.1 | 97.0 | 91.0 | 95.4 | 91.8 |
| 3ViewSense-4B-rl-strict (ours) | **95.0** | 88.2 | **98.7** | **93.3** | 97.4 | 93.2 |
| 3ViewSense-4B-rl-slack (ours) | 94.4 | **88.6** | **98.7** | 92.3 | **98.4** | **93.4** |

training and (ii) encourage the model to internalize teacher-generated view-grounded traces as its own reasoning process rather than merely imitating surface forms. Given a context $c$ (including $I_{\text{ego}}, q$, and $\hat{\mathcal{V}}$), we sample a group of $G$ completions $\{o_i\}_{i=1}^{G} \sim \pi_{\theta_{\text{old}}}(\cdot \mid c)$ and assign each completion a scalar verified reward $R_i$. We compute a group-normalized advantage $\hat{A}_i = (R_i - \mu_R)/(\sigma_R + \delta)$, where $\mu_R$ and $\sigma_R$ are the mean and standard deviation of $\{R_j\}_{j=1}^{G}$, and optimize the clipped objective:

$$\mathcal{J}_{\text{GRPO}}(\theta) = \mathbb{E}_{c, \{o_i\}} \Big[ \frac{1}{G} \sum_{i=1}^{G} \min \Big( r_i(\theta)\hat{A}_i, \ \text{clip}(r_i(\theta), 1-\epsilon, 1+\epsilon)\hat{A}_i \Big)$$
$$- \beta \, \mathbb{D}_{\text{KL}}(\pi_\theta \| \pi_{\text{ref}}) \Big].$$

(6)

where $r_i(\theta) = \pi_\theta(o_i \mid c)/\pi_{\theta_{\text{old}}}(o_i \mid c)$.

**Math-verified reward design.** For OrthoMind-3D, each ground-truth answer is either an integer count or a discrete relative direction. We consider two reward configurations. The strict reward uses exact-match verification, i.e., $R_{\text{strict}}(\hat{a}, a) = \mathbb{I}[\hat{a} = a]$, yielding $M_{\text{RL}}^{\text{strict}}$. The slack reward provides denser feedback for near-correct predictions, yielding $M_{\text{RL}}^{\text{slack}}$. For counting questions, given prediction $\hat{y}$ and ground truth $y$, we define $R_{\text{count}}(\hat{y}, y) = \max\{0, 1 - 0.2|\hat{y} - y|\}$, which assigns rewards $1, 0.8, 0.6, 0.4, 0.2$, and $0$ for absolute errors $0, 1, 2, 3, 4$, and $\geq 5$, respectively. For direction questions, given prediction $\hat{d}$ and ground truth $d$, we define $R_{\text{dir}}(\hat{d}, d) = 1$ if $\hat{d} = d$, $0.5$ if $\hat{d}$ and $d$ share at least one directional axis, and $0$ otherwise, where examples such as *left* and *front-left* are treated as partially aligned.

# 4. Experiments

## 4.1. Experimental Setup

We follow the training framework in Section 3.3 and train on the In-Domain split of OrthoMind-3D (Section 3.2). Out-of-Domain data are reserved for evaluation. We use Qwen3-VL-4B-Instruct (Bai et al., 2025) as the base model to evaluate the effectiveness and generalization of 3ViewSense and OrthoMind-3D.

**Stage I (OMS SFT).** We fine-tune the model to generate structured orthographic three-view descriptions from a single egocentric input. In total, we use 19.5k training instances and a detailed breakdown is reported in Appendix C.2.

**Stage II (VGR SFT).** We further fine-tune the model to perform view-grounded natural-language reasoning conditioned on the three views and output the final answer. By comparing the correctness of the answers produced along the reasoning chains, we filter the samples and finally obtain 21k training instances. The reasoning traces are generated by Gemini-3-Flash using the prompt in Appendix 9.

**RL refinement (GRPO).** For the RL variants reported in our main results, we further optimize the Stage II model with GRPO as described in Section 3.3. We use 30k RL instances, including 10k re-sampled from the Stage II pool and 20k newly generated instances. We report two reward settings, strict and slack, matching the definitions in Section 3.3.

## 4.2. Evaluation

**Evaluation datasets.** We evaluate on OrthoMind-3D and several public benchmarks. OrthoMind-3D is split into In-Domain (ID) and Out-of-Domain (OOD) subsets (Section 3.2). For ID evaluation, we strictly de-duplicate against all training data and then sample a disjoint held-out test set.

We cover two task families (Block Counting and Object Reasoning), and report both cardinality queries and attribute-conditioned queries (e.g., color). Appendix Table 9 summarizes the split composition.

**Additional benchmarks.** We further evaluate on several external spatial reasoning benchmarks, including the full MindCube-Tiny subset from MindCube (Yin et al., 2025) (1,040 examples), CV-Bench 2D (Tong et al., 2024) (1,438 examples), the counting and relative-position subsets of SPBench-SI (Li et al., 2025c) (306 examples), the Ego-centric split of OmniSpatial Perspective-Taking (Jia et al., 2026) (102 examples), and the Camera Perspective split of ViewSpatial (Li et al., 2025b) (2,769 examples). Details of the filtering and subsampling procedure are provided in the Appendix C.1.

**Metric and protocol.** We report (pass@1) accuracy under identical decoding settings across models, using exact-match after normalization for both integer counting and discrete direction labels. We compare against proprietary models, specialized spatial reasoning models, and open-source VLMs.

## 5. Results & Analysis

### 5.1. Main Results

**In-domain results on OrthoMind-3D.** We first evaluate models on the in-domain split (Table 1) across three groups: proprietary models, specialized spatial reasoning models, and open-source VLMs. Even on the seemingly simple block counting task, most models perform poorly; moreover, cardinality-only counting is substantially harder than attribute-conditioned counting, likely because salient attributes (e.g., color) turn the problem into easier attribute-specific counting rather than true 3D enumeration. Our 3ViewSense model improves over open-source baselines after SFT but remains imperfect on counting, while GRPO refinement with both strict and slack rewards further lifts in-domain performance to consistently high accuracy across tasks, which is expected given the simplicity of OrthoMind-3D queries.

**OOD generalization and transfer to external benchmarks.** We further evaluate on the OrthoMind-3D OOD split and other spatial benchmarks (Table 2). 3ViewSense exhibits generalization at a similar parameter scale, with clearer gains on block counting and object positioning. RL refinement further improves over the SFT-only model on OOD tasks, and the slack reward generally performs better than the strict reward. On external benchmarks, the gains transfer to all five benchmarks, suggesting that the learned three-view reasoning ability is not limited to OrthoMind-3D.

### 5.2. In-depth Analysis

**In-Context Learning Analysis.** We ask whether a model can acquire three-view mental reasoning purely from a few in-context demonstrations, without any parameter updates. Using the few-shot instruction and teaching examples in Appendix B.2, we evaluate multiple models on the OrthoMind-3D in-domain test set; results are summarized in Figure 4 (A). We observe that only the strongest proprietary models show limited improvements under ICL, while most open-source VLMs degrade. This suggests that three-view reasoning is not merely a prompt-following skill: without an internalized view-consistent representation, the model struggles to reliably translate egocentric visual cues into orthographic constraints, and the additional steps introduced by ICL may amplify rather than reduce such misalignment.

**Explicit Three-View Description Analysis.** Next, we test whether providing an explicit orthographic three-view description can directly improve spatial reasoning, as shown in Figure 4 (B). Across models, injecting the three-view description yields substantial gains, especially on occlusion-heavy block counting, indicating that many models possess sufficient symbolic capacity once the spatial structure is exposed through a view-consistent interface. While the effectiveness varies across models and tasks, these differences suggest that the benefit of external spatial cues depends on a model's ability to coherently integrate multi-view information. Overall, the results support our core insight: the primary bottleneck lies in the absence of a stable intermediate spatial representation, motivating 3ViewSense to explicitly learn to induce and integrate orthographic views rather than relying on prompt-only adaptation or external annotations at inference time.

**Model Response Analysis.** Table 3 shows that the base model exhibits extreme verbosity on block counting (e.g., > 10k tokens), suggesting that, without explicit spatial representation, the model tends to revisit uncertain spatial hypotheses, which can cause drift and hallucinated intermediate states and eventually hurt accuracy. In contrast, 3ViewSense consistently produces concise outputs by guiding the model to reason from an engineering-inspired three-view perspective: it first forms view-consistent orthographic mental sketches and then composes the final answer, reducing ambiguity and redundant deliberation. This qualitative behavior is further illustrated in Appendix C.6 (Figure 13).

*Table 2.* OOD generalization on OrthoMind-3D and comparison on external benchmarks. We report accuracy (%) under the same evaluation protocol as Section 4.2. ↑ indicates relative improvement over the Qwen3-VL-4B-Instruct base model; ↓ indicates relative drop.

| Model | (A) OrthoMind-3D (OOD Task) | | | | (B) Other Spatial Benchmarks | | | | |
|---|---|---|---|---|---|---|---|---|---|
| | Block Count | Block Count (Attr.) | Object Count | Object Position | MindCube-Tiny | CV-Bench (2D) | OmniSpatial | SPBench (SI) | ViewSpatial |
| SpatialLadder-3B | 19.6 | 43.0 | 22.2 | 15.6 | 38.5 | 68.6 | 67.6 | **30.8** | 36.2 |
| SpaceOm-4B | 23.4 | 61.7 | 46.2 | 19.2 | 35.5 | 68.8 | 67.6 | 29.7 | 34.8 |
| SpaceQwen2.5-VL-3B-Instruct | 21.2 | 62.5 | 51.8 | 16.5 | 36.4 | 66.1 | 58.8 | 28.1 | 30.1 |
| InternVL3.5-4B | 33.1 | 65.9 | 54.6 | 34.8 | 14.4 | 79.4 | 67.7 | 28.4 | 34.1 |
| Qwen2.5-VL-3B | 26.8 | 58.7 | 47.2 | 20.1 | 36.1 | 68.7 | 61.7 | 27.1 | 33.4 |
| Qwen3-VL-4B-Instruct | 21.2 | 57.8 | 50.9 | 46.7 | 27.2 | 77.9 | 58.1 | 22.2 | 35.5 |
| Qwen3-VL-8B-Instruct | 25.5 | 64.7 | **56.5** | 55.0 | 36.2 | **81.1** | **73.5** | 22.8 | **38.7** |
| 3ViewSense-4B-sft (ours) | 31.1 (↑46.7%) | 62.1 (↑7.4%) | 32.4 (↓36.3%) | 72.5 (↑55.2%) | 34.9 (↑28.3%) | 74.3 (↓4.6%) | 56.7 (↓2.4%) | 20.6 (↓7.2%) | 34.4 (↓3.1%) |
| 3ViewSense-4B-rl-strict (ours) | 33.2 (↑56.6%) | **71.1** (↑23.0%) | 49.1 (↓3.5%) | 74.3 (↑59.1%) | 36.7 (↑34.9%) | 78.1 (↑0.3%) | 59.1 (↑1.7%) | 23.2 (↑4.5%) | 36.6 (↑3.1%) |
| 3ViewSense-4B-rl-slack (ours) | **38.7** (↑82.5%) | 70.2 (↑21.5%) | 50.9 (0.0%) | **76.1** (↑63.0%) | **38.9** (↑43.0%) | 79.9 (↑2.6%) | 62.8 (↑8.1%) | 25.4 (↑14.4%) | 37.1 (↑4.5%) |

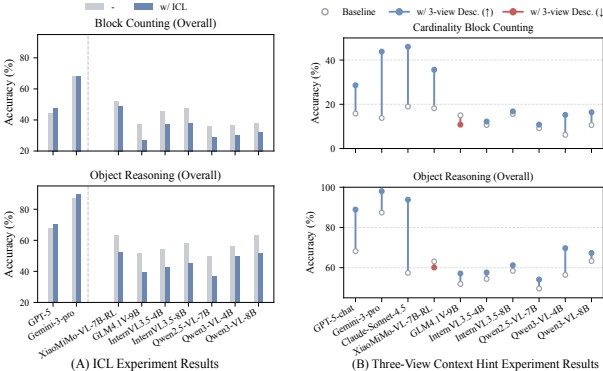

(A) ICL Experiment Results     (B) Three-View Context Hint Experiment Results

*Figure 4.* In-context learning (ICL) and explicit orthographic three-view description study on OrthoMind-3D (in-domain). ICL yields limited improvements only for the strongest proprietary models, while explicit three-view descriptions substantially improve performance for most models, supporting the need for a view-consistent intermediate representation.

*Table 3.* Average response length (tokens) across benchmarks. For this simple task, the base model tends to produce overly verbose reasoning (over thinking) and can even suffer reduced accuracy; 3ViewSense substantially reduces this redundancy.

| Model | Block Count | Block Count (Attr.) | SPBench-SI | ViewSpatial |
|---|---|---|---|---|
| Qwen3-VL-4B-Instruct | 10218.9 | 6531.1 | 451.6 | 952.1 |
| 3ViewSense-4B-sft | 350.4 | 375.2 | 250.1 | 261.7 |
| 3ViewSense-4B-rl-strict | 377.1 | 378.9 | 273.5 | 266.6 |
| 3ViewSense-4B-rl-slack | 375.2 | 389.4 | 281.5 | 270.9 |

## 5.3. Ablation Study

**Direct QA vs. 3ViewSense reasoning.** We examine whether the gains come merely from supervised data or from the proposed view-guided reasoning process. Specifically, we construct a Direct QA variant with the same training inputs, instances, and hyperparameters as VGR-SFT, but use only the final answer as the target output. This removes intermediate view-guided reasoning supervision while preserving answer-level supervision. As shown in Table 4, Direct QA performs better on OrthoMind-3D, suggesting stronger fitting to the target dataset. However, its performance drops sharply on external benchmarks, achieving only 1.3 on SPBench-SI and 7.2 on ViewSpatial. In

*Table 4.* Ablation on the SFT target. Direct QA uses the same inputs, instances, and hyperparameters as VGR-SFT, but supervises only the final answer. We report average accuracy (%) on OrthoMind-3D (In-Domain and Out-of-Domain) and two external benchmarks (SPBench-SI and ViewSpatial).

| SFT Target | OrthoMind-3D | | SPBench-SI | ViewSpatial |
|---|---|---|---|---|
| | ID | OOD | | |
| Direct QA | **80.3** | **49.8** | 1.3 | 7.2 |
| 3ViewSense Reasoning | 70.3 | 46.6 | **21.2** | **34.1** |

*Table 5.* Ablation study on the two-stage SFT design. We report accuracy (%) on OrthoMind-3D (In-Domain and Out-of-Domain) and two external benchmarks (MindCube-Tiny and ViewSpatial).

| Stage | OrthoMind-3D | | MindCube-Tiny | ViewSpatial |
|---|---|---|---|---|
| | ID | OOD | | |
| OMS-SFT (only) | 48.7 | 41.3 | 29.6 | 33.4 |
| VGR-SFT (only) | 70.3 | 46.6 | 32.4 | 34.1 |
| Two-stage SFT (OMS→VGR) | **78.6** | **49.5** | **34.9** | **34.4** |

contrast, 3ViewSense reasoning generalizes substantially better across benchmarks. These results suggest that the improvement is not solely attributable to supervised data; the view-guided reasoning process plays an important role in transferable spatial reasoning.

**SFT stage ablation.** Table 5 evaluates the two-stage SFT design. OMS-SFT alone yields limited performance, indicating that orthographic sketch induction by itself is insufficient for downstream spatial queries. VGR-SFT brings substantial gains by learning to integrate the induced views for answer prediction. The full two-stage SFT further improves over VGR-SFT alone across all benchmarks, especially on OrthoMind-3D, suggesting that OMS-SFT provides a useful view-induction initialization that complements answer-oriented VGR training.

**RL stage analysis.** Figure 5 shows the cumulative reward curves during GRPO training under two initialization settings. Directly launching RL from the Stage I OMS-SFT checkpoint leads to high-variance reward oscillations without a sustained upward trend, reflecting unstable, format- or heuristic-driven exploration that often results in training collapse and degraded downstream performance. In con-

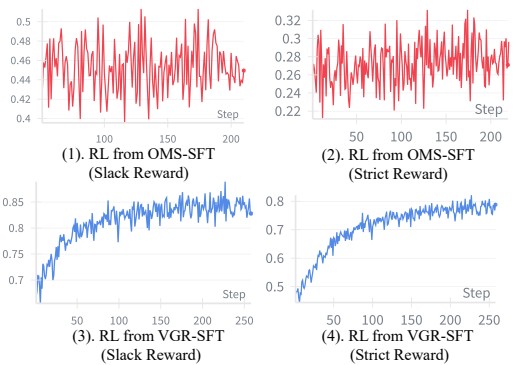

*Figure 5.* RL ablation on initialization. We compare GRPO reward trajectories when starting RL from the Stage I OMS-SFT model versus from the Stage II VGR-SFT model.

trast, initializing from the Stage II VGR-SFT model yields a steadily increasing reward with substantially reduced variance, indicating that view-grounded SFT provides a critical inductive bias: the policy already conditions on inferred orthographic views and produces verifiable intermediate reasoning, allowing RL to consistently reinforce correct spatial behavior. This ablation shows that while OMS SFT is important, a dedicated view-grounded warm-start is essential for stable and effective RL optimization.

## 6. Limitation and Future Work

Our core insight is that many VLM failures on spatial reasoning stem from the lack of a view-consistent intermediate representation, and our contributions are OrthoMind-3D for diagnosing these failure modes and 3ViewSense for learning a simulate-and-reason pipeline that induces orthographic views for view-grounded reasoning. The main limitation is that not all spatial reasoning problems are well captured by three orthographic views alone, since many tasks require additional physical and semantic priors (e.g., support relations, affordances, and dynamics) beyond geometry. Future work will focus on learning mechanisms that estimate view-induction uncertainty and adaptively decide when to invoke orthographic mental simulation, expanding the intermediate representation beyond three fixed views via task-dependent or hybrid spatial abstractions, and integrating view-grounded reasoning into larger general-purpose multimodal models with continual learning to preserve the induced reasoning behavior while reducing catastrophic forgetting.

## 7. Conclusion

Vision-language models often fail on spatial tasks like block counting under occlusion, suggesting a spatial intelligence gap caused by the lack of a stable, view-consistent repre-

sentation. We propose 3ViewSense, a simulate-and-reason framework that induces orthographic views and performs view-grounded reasoning, and we introduce OrthoMind-3D to diagnose occlusion-heavy counting and object reasoning. Across in-domain, out-of-domain, and external benchmarks, 3ViewSense yields accuracy gains and better robustness and conciseness, with additional improvements from RL refinement. Nonetheless, orthographic mental simulation is not universally sufficient, and performance can degrade when view induction is unreliable or when tasks require richer priors beyond geometric projections. Future work will extend view-consistent reasoning to more open-world scenes and more complex interactions, and explore how models can adaptively choose such structured representations.

## Impact Statement

This work aims to advance spatial reasoning in vision-language models through a diagnostic benchmark and a view-grounded reasoning framework. Potential positive impacts include improving the reliability of multimodal systems in applications that require spatial understanding, such as embodied AI, robotics, assistive perception, and visual question answering. Potential risks arise when stronger spatial reasoning models are deployed in safety-sensitive physical environments, where incorrect predictions may affect downstream actions or human decisions. Our benchmark uses synthetic and controlled scenes, which reduces direct privacy concerns, but it does not eliminate the need for careful evaluation before open-world deployment.

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

## A. Preliminary Analysis

### A.1. Uniqueness Conditions for Three-View Counting

**Definition A.1** (Notation). For a $W \times L$ grid, let $H_{x,y} \in \mathbb{N}$ be the height at position $(x, y)$. Define:

$$M_x^c = \max_{y'} H_{x,y'} \quad \text{(front view)},$$

$$M_y^r = \max_{x'} H_{x',y} \quad \text{(side view)},$$

$$O_{x,y}^c = \max_{y' \neq y} H_{x,y'} \quad \text{(others in same column)},$$

$$O_{x,y}^r = \max_{x' \neq x} H_{x',y} \quad \text{(others in same row)}.$$

The three views are $\mathcal{V}(H) = (\{H_{x,y} > 0\}, \{M_x^c\}, \{M_y^r\})$.

**Theorem A.2** (Uniqueness). *A configuration $H$ is uniquely determined by $\mathcal{V}(H)$ if and only if for every $(x, y)$ with $H_{x,y} > 0$,*

$$(H_{x,y} = 1 \wedge (M_x^c = 1 \vee M_y^r = 1)) \vee (H_{x,y} > 1 \wedge (H_{x,y} > O_{x,y}^c \vee H_{x,y} > O_{x,y}^r)). \tag{7}$$

*Proof.* **1. Sufficiency ($\Rightarrow$).** Assume Eq.(7) holds. For any $(x, y)$ with $H_{x,y} > 0$:

- If $H_{x,y} = 1$ and $M_x^c = 1$, then all nonzero cells in column $x$ must be 1. Thus $H_{x,y}$ is forced to 1. The case $M_y^r = 1$ is symmetric.

- If $H_{x,y} > 1$ and $H_{x,y} > O_{x,y}^c$, then $H_{x,y}$ is the unique maximum in its column, so $H_{x,y} = M_x^c$. The case $H_{x,y} > O_{x,y}^r$ symmetrically forces $H_{x,y} = M_y^r$.

Thus each $H_{x,y}$ is uniquely determined by the views, implying uniqueness of the entire configuration.

**2. Necessity ($\Leftarrow$).** We prove the contrapositive. Suppose there exists $(x_0, y_0)$ violating Eq.(7). Then $H_{x_0,y_0} > 0$ and:

- Either $H_{x_0,y_0} = 1$ with $M_{x_0}^c > 1$ and $M_{y_0}^r > 1$, or

- $H_{x_0,y_0} > 1$ with $H_{x_0,y_0} \leq O_{x_0,y_0}^c$ and $H_{x_0,y_0} \leq O_{x_0,y_0}^r$.

In both cases, we can construct an alternative configuration $H'$ with the same views:

- In the first case, choose $y_1$ such that $H_{x_0,y_1} = M_{x_0}^c > 1$. Let $H'_{x_0,y_0} = 0$, $H'_{x_0,y_1} = M_{x_0}^c$, and keep other heights unchanged. Adjustments can be made to maintain $M_{y_0}^r$ (e.g., by increasing another cell in row $y_0$).

- In the second case, since $H_{x_0,y_0}$ is not a unique maximum in its row or column, we can decrease it by 1 and increase another cell in the same row/column without changing $M_{x_0}^c$ or $M_{y_0}^r$.

In either construction, $\mathcal{V}(H') = \mathcal{V}(H)$ but $H' \neq H$, contradicting uniqueness. $\square$

**Corollary A.3.** *If condition Eq.(7) holds, the total block count $\sum_{x,y} H_{x,y}$ is uniquely determined by the three views.*

## B. Additional Method Details

### B.1. Datasets Details

This section provides qualitative examples and annotation formats for OrthoMind-3D. Figure 6 shows representative samples from the in-domain and out-of-domain subsets, covering both block counting and object reasoning. Figure 7 illustrates the orthographic three-view description format and the Stage-I (OMS) instruction used for supervised fine-tuning.

## B.2. Prompt Template

This section summarizes the prompt templates used in our data generation and evaluation. Figure 8 presents the template for synthesizing photorealistic multi-object scenes with controllable spatial layouts. Figure 9 provides the template for converting orthographic three-view descriptions into natural-language reasoning traces for Stage-II supervision. Figure 10 and Figure 11 list the instructions and teaching demonstrations used in the in-context learning study.

## C. Additional Experiments Details

### C.1. Benchmark Subsampling Procedure

Our main experiments focus on single-image, egocentric spatial queries. To ensure comparability across public benchmarks with heterogeneous task definitions, we apply a unified filtering and sampling procedure and report the resulting instance counts.

**MindCube.** We use the full **MindCube-Tiny** split without additional subsampling, resulting in 1,040 instances.

**CV-Bench.** We restrict CV-Bench to its 2D category (denoted as CV-Bench 2D) and evaluate on all 1,438 instances in this subset.

**SPBench.** We use the SPBench-SI split (single-image input) and keep only questions whose answer types match our setting: counting and relative-position classification. This yields 306 instances.

**OmniSpatial.** OmniSpatial contains four tasks; we use only Perspective_Taking and further restrict to the Egocentric subset, resulting in 102 instances.

**ViewSpatial.** We evaluate on the Camera perspective category and include all 2,769 instances in this subset.

### C.2. Experiments Settings

**Reproducibility.** We report the key hyperparameters for our two-stage training pipeline, where both Orthographic Mental Simulation (OMS) and View-Grounded Reasoning (VGR) are trained with supervised fine-tuning (SFT), followed by GRPO-based reinforcement learning for refining the VGR stage. We use LLaMA-Factory (Zheng et al., 2024) for SFT (OMS and VGR) and verl (Sheng et al., 2025) for the GRPO refinement. Table 6 summarizes the hyperparameter settings.

*Table 6.* Training hyperparameters for SFT and RL (GRPO, Stage II refinement).

| SFT (Training Hyperparameters) | | RL (Training Hyperparameters) | |
|---|---|---|---|
| Hyperparameter | Value | Hyperparameter | Value |
| per_device_train_batch_size | 1 | RL algorithm | GRPO |
| gradient_accumulation_steps | 8 | Training epochs | 5 |
| bf16 | true | Train batch size | 512 |
| data_seed | 42 | Actor learning rate | $1.0 \times 10^{-6}$ |
| gradient_checkpointing | true | Max prompt length | 8192 |
| lr_scheduler_type | cosine | Max response length | 16384 |
| warmup_ratio | 0.1 | Rollout samples / prompt | 8 |
| num_train_epochs | 1 | KL regularization | enabled ($\beta = 0.01$) |
| max_pixels | 262144 | Reward function | custom (slack / strict) |
| min_pixels | 1024 | | |
| deepspeed | stage3 | | |

Stage II reasoning traces are generated by Gemini-3-Flash using the prompt template in Appendix 9. We report two RL variants using strict and slack reward settings (Section 3.3).

*Table 7.* Training data statistics and splits for OMS SFT (Stage I).

| Stage | Total | Block count | Distinct block count | Object reasoning | Distinct object reasoning |
|---|---|---|---|---|---|
| OMS SFT (Stage I) | 19,544 | 6,000 | 5,000 | 3,544 | 5,000 |

Stage II (VGR SFT) data are sampled from the Stage I pool.

*Table 8.* Additional training data statistics.

| Stage | Total |
|---|---|
| VGR SFT (Stage II) | 21,000 |
| RL (GRPO) | 30,000 (10,000 from Stage II + 20,000 new) |

**OrthoMind-3D evaluation split.** Table 9 reports the instance counts for each evaluation split (In-Domain vs. Out-of-Domain) and task category. Attribute-conditioned queries specify an explicit attribute (e.g., color), while cardinality queries evaluate pure counting or spatial reasoning without attribute cues.

*Table 9.* OrthoMind-3D evaluation split statistics. "Attribute-conditioned" denotes queries that specify explicit attributes (e.g., color).

| Split | Task | # |
|---|---|---|
| ID | Block counting (cardinality) | 500 |
| ID | Block counting (attribute-cond.) | 2,202 |
| ID | Object reasoning: counting | 300 |
| ID | Object reasoning: positioning | 300 |
| ID | Object reasoning: counting (attr.) | 500 |
| ID | Object reasoning: positioning (attr.) | 500 |
| OOD | Block counting (cardinality) | 235 |
| OOD | Block counting (attribute-cond.) | 235 |
| OOD | Object reasoning: counting | 108 |
| OOD | Object reasoning: positioning | 109 |

### C.3. Experiment Details for Visual Information Sufficiency

Multimodal Large Language Models (VLMs) frequently struggle with geometric reasoning tasks. We conducted a diagnostic probing experiment to determine whether these errors stem from the visual encoder's inability to capture spatial information or the language model's failure to utilize these features. We hypothesize that if a lightweight classifier is capable of achieving high accuracy using frozen visual features, this result would demonstrate that the visual encoder has already extracted sufficient information. Consequently, the failure must stem from the downstream reasoning process. Through this probing analysis, we can therefore explicitly identify the locus of the performance bottleneck.

**Experimental Setup.** We utilize the Block Counting dataset in OrthoMind-3D to predict the total number of stacked cubes. The ground truth labels $y$ lie in the range $\mathcal{Y}_{gt} = \{2, \ldots, 38\}$. We selected Qwen3-VL-4B-Instruct as the base model. Formally, let $\mathcal{E}_v(\cdot)$ denote the frozen visual encoder of the VLM. For each input image $x$, we extract the visual representation vector $\mathbf{z} \in \mathbb{R}^d$:

$$\mathbf{z} = \mathcal{E}_v(x), \quad \text{where } d = 2560. \tag{8}$$

We then train a lightweight Multi-Layer Perceptron (MLP) probe, denoted as $f_\theta : \mathbb{R}^d \to \mathbb{R}^C$, parameterized by $\theta$. To simulate a realistic setting where the probe is agnostic to the specific upper bound of the dataset, we set the number of output classes to $C = 50$, which is a superset of the ground truth label space ($|\mathcal{Y}_{gt}| < C$). The probe is trained to minimize the standard cross-entropy loss $\mathcal{L}_{CE}$ on the training set $\mathcal{D}_{train}$:

$$\theta^* = \arg\min_\theta \sum_{(x_i, y_i) \in \mathcal{D}_{train}} \mathcal{L}_{CE}(f_\theta(\mathcal{E}_v(x_i)), y_i). \tag{9}$$

We experimented with MLP depths ranging from 2 to 4 layers to evaluate the linear separability and non-linear information content of $\mathbf{z}$.

**Results and Analysis.** The classification accuracy of the probes on the test split is reported in Table 10. Simple MLP probes achieved remarkable accuracy. The 4-layer MLP reached 55.8% and substantially outperformed current state-of-the-art VLMs on this task. Even the most lightweight 2-layer MLP achieved 43.2%. Additionally, we observe that accuracy improves consistently as the probe depth increases. This trend suggests that the geometric information is implicitly encoded in the visual features and requires non-linear transformations to be decoded effectively. Crucially, even the deepest 4-layer probe is negligible in size compared to the massive parameter count of the LLM.

*Table 10.* Block Counting Probe Accuracy on Qwen3-VL Visual Features.

| Probe Architecture | Layers | Accuracy |
|---|---|---|
| MLP [512, 256] | 2 | 43.2% |
| MLP [1024, 512, 256] | 3 | 50.4% |
| MLP [1024, 512, 256, 128] | 4 | 55.8% |

This significant performance gap provides compelling evidence that the visual encoder is not a "visual blind spot." Instead, it demonstrates that the encoder successfully extracts fine-grained quantitative features sufficient for this task. Consequently, it indicates that the LLM fails to correctly align or utilize these available visual features during its reasoning process. This finding justifies the necessity of our proposed method to bridge this reasoning gap.

### C.4. Experiment Results for 3-view Description Reasoning

Table 11 studies whether providing an explicit orthographic three-view description (front/left/top) can directly improve spatial reasoning accuracy. We report the original single-view setting ("–") and the setting augmented with a three-view description ("w/ 3-view Desc.").

*Table 11.* Effect of explicit orthographic three-view descriptions on OrthoMind-3D. We report accuracy (%) for Cardinality Block Counting and Object Reasoning (Overall). Colored numbers indicate the relative change compared to the single-view baseline (green: improvement; red: drop).

| Model | Cardinality Block Counting | | Object Reasoning (Overall) | |
|---|---|---|---|---|
| | – | w/ 3-view Desc. (Rel.↑) | – | w/ 3-view Desc. (Rel.↑) |
| GPT-5 | 15.8 | 28.6 (+81.0%) | 68.2 | 88.9 (+30.4%) |
| Gemini-3-pro | 13.8 | 43.8 (+217.4%) | 87.4 | 98.0 (+12.1%) |
| Claude-Sonnet-4.5 | 19.0 | 46.0 (+142.1%) | 57.4 | 93.8 (+63.4%) |
| XiaoMiMo-VL-7B-RL | 18.2 | 35.6 (+95.6%) | 63.1 | 60.1 (-4.7%) |
| GLM4.1V-9B | 15.0 | 10.8 (-28.0%) | 51.9 | 57.1 (+10.0%) |
| InternVL3.5-4B | 10.6 | 12.2 (+15.1%) | 54.4 | 57.6 (+5.9%) |
| InternVL3.5-8B | 15.6 | 16.8 (+7.7%) | 58.4 | 61.2 (+4.8%) |
| Qwen2.5-VL-7B | 9.2 | 10.8 (+17.4%) | 49.6 | 54.1 (+9.1%) |
| Qwen3-VL-4B-Instruct | 6.2 | 15.2 (+145.2%) | 56.4 | 69.7 (+23.6%) |
| Qwen3-VL-8B-Instruct | 10.6 | 16.4 (+54.7%) | 63.3 | 67.3 (+6.3%) |

Overall, adding a three-view description improves most models, with the largest gains on occlusion-heavy block counting. Proprietary models benefit most (e.g., Gemini-3-pro: +217.4%), suggesting that a structured, view-consistent representation is often the key bottleneck. Open-source VLMs show mixed results: while many improve, some degrade (e.g., GLM4.1V-9B on block counting; XiaoMiMo-VL-7B-RL on object reasoning), implying that extra views can introduce noise when fusion is weak. These results motivate 3ViewSense, which internalizes orthographic-view abstraction and learns to fuse views into a coherent 3D mental model.

### C.5. Analysis of the cumulative reward curve in the RL stage

Figure 12 shows the cumulative reward trajectory during GRPO-based RL, initialized from the Stage II VGR SFT model. Both strict and slack reward settings exhibit a consistent upward trend without sustained oscillation or collapse, suggesting stable policy improvement under math-verifiable supervision. Compared to the strict reward, the slack reward typically yields smoother training dynamics due to denser partial-credit feedback (Section 3.3), while the strict reward can be higher-variance because any small prediction error leads to zero reward. Overall, these curves indicate that RL effectively refines answer correctness while preserving the view-grounded reasoning behavior learned during SFT.

### C.6. Qualitative Model Response Examples

We qualitatively compare responses from the base model and 3ViewSense on occlusion-heavy block counting. As shown in Figure 13, the base model often produces lengthy and repetitive reasoning with unstable intermediate hypotheses, whereas 3ViewSense yields more concise and structured traces by explicitly grounding the reasoning process in orthographic-view

mental simulation.

## C.7. Case Study on External Benchmarks

Figure 14 presents qualitative examples of 3ViewSense on ViewSpatial and SPBench-SI. Although these benchmarks differ from OrthoMind-3D in task format and scene distribution, 3ViewSense still follows a view-grounded reasoning process: it identifies the relevant spatial configuration, forms intermediate view-consistent observations, and derives the final answer from them. These examples provide qualitative evidence that the learned reasoning behavior can transfer to external spatial reasoning benchmarks.

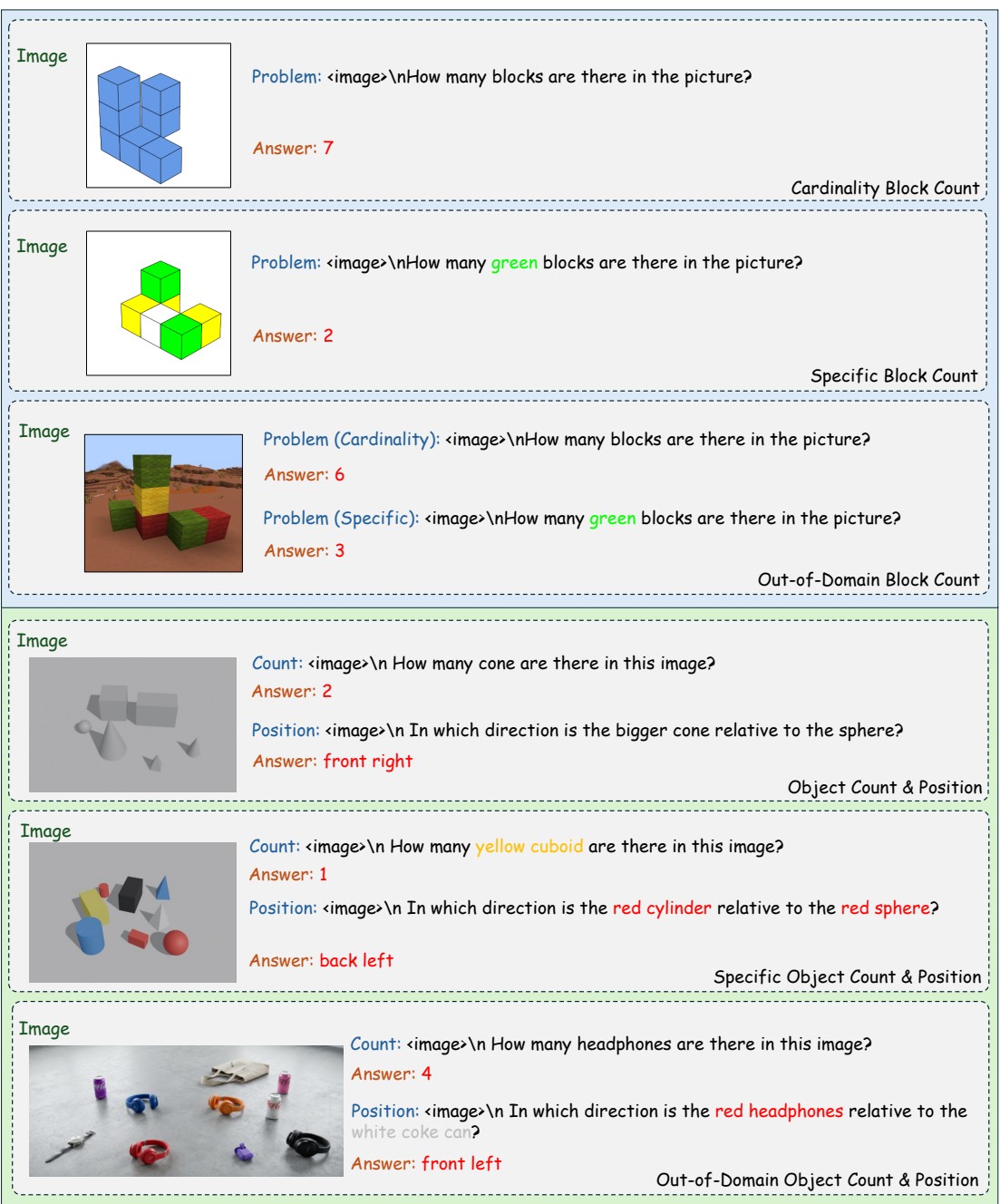

*Figure 6.* Sample visualization of the OrthoMind-3D dataset. We show examples from the in-domain and out-of-domain subsets for two task families: block counting (top) and object reasoning (bottom). In-domain data are generated with strict geometric constraints, while out-of-domain data are photorealistic and less structured to evaluate generalization.

**Instruction for Stage1 SFT (Block Counting):** <image>\nFirst, carefully inspect the blocks in the image to understand the overall structure and count. Then analyze the 2D projections from three viewpoints: Front, Left, and Top. Finally, output a JSON object that provides the integer-grid 2D coordinates of the blocks in each view.\n {\"front-view\":[{\"x\":int,\"y\":int,\"z\":int}],\"left-view\":[{\"x\":int,\"y\":int,\"z\":int}],\"top-view\":[{\"x\":int,\"y\":int,\"z\":int}]}"

**Instruction for Stage1 SFT (Object Reasoning):** <image>\nFirst, scan the image to identify and verify all geometric objects present (candidate set: bigger red square pyramid, smaller green square pyramid, bigger black square pyramid, bigger blue square pyramid, smaller blue square pyramid, bigger yellow square pyramid, bigger yellow cone, smaller black cone), and understand their counts and relative spatial relationships. Then analyze the scene from three viewpoints: front, left, and top. In the front view, list the visible objects ordered from left to right; in the left view, list the visible objects ordered from back to front; in the top view, provide two ordered lists: from left to right and from back to front. Output only one JSON object that follows the schema below, with no additional text or explanation.\n {\"front-view\":{\"from-left-to-right\":List},\"left-view\":{\"from-back-to-front\":List},\"top-view\":{\"from-left-to-right\":List,\"from-back-to-front\":List}}

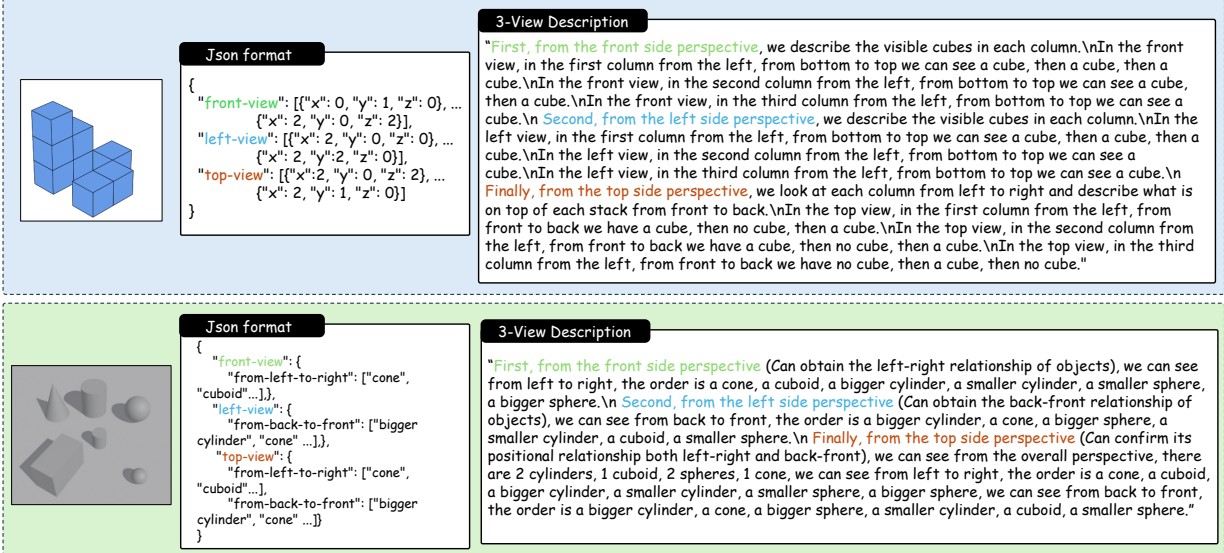

**Json format**

```
{
  "front-view": [{"x": 0, "y": 1, "z": 0}, ...
                 {"x": 2, "y": 0, "z": 2}],
  "left-view": [{"x": 2, "y": 0, "z": 0}, ...
                {"x": 2, "y":2, "z": 0}],
  "top-view": [{"x":2, "y": 0, "z": 2}, ...
               {"x": 2, "y": 1, "z": 0}]
}
```

**3-View Description**

"First, from the front side perspective, we describe the visible cubes in each column.\nIn the front view, in the first column from the left, from bottom to top we can see a cube, then a cube, then a cube.\nIn the front view, in the second column from the left, from bottom to top we can see a cube, then a cube.\nIn the front view, in the third column from the left, from bottom to top we can see a cube.\n Second, from the left side perspective, we describe the visible cubes in each column.\nIn the left view, in the first column from the left, from bottom to top we can see a cube, then a cube, then a cube.\nIn the left view, in the second column from the left, from bottom to top we can see a cube.\nIn the left view, in the third column from the left, from bottom to top we can see a cube.\n Finally, from the top side perspective, we look at each column from left to right and describe what is on top of each stack from front to back.\nIn the top view, in the first column from the left, front to back we have a cube, then no cube, then a cube.\nIn the top view, in the second column from the left, from front to back we have a cube, then no cube, then a cube.\nIn the top view, in the third column from the left, from front to back we have no cube, then a cube, then no cube."

**Json format**

```
{
  "front-view": {
    "from-left-to-right": ["cone",
"cuboid"...],},
  "left-view": {
    "from-back-to-front": ["bigger
cylinder", "cone" ...],},
  "top-view": {
    "from-left-to-right": ["cone",
"cuboid"...],
    "from-back-to-front": ["bigger
cylinder", "cone" ...]}
}
```

**3-View Description**

"First, from the front side perspective (Can obtain the left-right relationship of objects), we can see from left to right, the order is a cone, a cuboid, a bigger cylinder, a smaller cylinder, a smaller sphere, a bigger sphere.\n Second, from the left side perspective (Can obtain the back-front relationship of objects), we can see from back to front, the order is a bigger cylinder, a cone, a bigger sphere, a smaller cylinder, a cuboid, a smaller sphere.\n Finally, from the top side perspective (Can confirm its positional relationship both left-right and back-front), we can see from the overall perspective, there are 2 cylinders, 1 cuboid, 2 spheres, 1 cone, we can see from left to right, the order is a cone, a cuboid, a bigger cylinder, a smaller cylinder, a smaller sphere, a bigger sphere, we can see from back to front, the order is a bigger cylinder, a cone, a bigger sphere, a smaller cylinder, a cuboid, a smaller sphere."

*Figure 7.* Example of the orthographic three-view description and the Stage-I (OMS) instruction used for training.

## A. Prompt for Generative AI-based Randomized Spatial Object Placement

**1. Systematic Prompt Template**

A wide-angle photorealistic 3D render of **[Selected Objects]** haphazardly scattered across a **[Selected Environment]**. The scene features significant negative space around objects. Items are placed on the same horizontal plane (floor) but with random orientations. Each object is strictly isolated, no overlapping, no touching, widely spaced apart. High angle shot, depth of field, soft volumetric lighting, Octane render, 8k resolution, ultra-detailed.

**2. Definition of Hyperparameters**

- **[Selected Objects]**: A randomized set of $N$ items (where $2 \leq N \leq 6$) selected from a predefined library. Each item can be further modified by:

  - *Quantity*: Numerical count and pluralization (e.g., "3 oranges").
  - *Visual Attributes*: Randomized modifiers including color (red, purple, etc.), scale (tiny, giant), and material state (matte, shiny, dusty, damaged).
  - *Uniqueness Constraint*: If required, a rule is appended: "Every single object must be distinct in visual appearance to be easily identifiable."

- **[Selected Environment]**: The background setting and floor texture, chosen from:

  - Modern living room (light wood texture)
  - Clean white gallery floor
  - Minimalist concrete studio
  - Marble hall surface
  - Open wooden deck

**3. Spatial Configuration Rules**

- **Ground-Plane Constraint**: All objects are anchored to the same vertical coordinate ($z$-axis) to ensure they rest naturally on the surface.

- **Non-Occlusion Principle**: Objects are distributed with a focus on "negative space," ensuring that each object's silhouette is fully visible from a high-angle perspective.

- **Stochastic Orientation**: Objects are rotated randomly along their local axes to simulate a non-artificial, "haphazard" distribution.

*Figure 8.* Prompt design for image generation using generative AI models.

## B. Prompt for Conversion from Three-View to Natural Language Reasoning

**System Prompt**
*You are a spatial reasoner who mentally reconstructs a 3D scene from partial viewpoints. You do not describe data; you form an internal 3D mental image and reason within it.*

**Reasoning Generation Prompt**

### ## Role
You are a spatial reasoner who directly perceives a 3D scene. You do not read data or analyze formats; you mentally see a solid block structure.

### ## Task
Given a Problem and Three views (front, left, top), write a single continuous reasoning narrative that reconstructs the scene, solves the problem, and ends with the final answer in \boxed{}.

### ## How to read the views
The views may appear in two forms.
1) **Block list format**
Each view lists blocks like '{x,y,z,color,visible}'. Interpret fields as spatial cues:
 • 'x': larger values correspond to positions further to the left
 • 'y': larger values correspond to positions further to the front
 • 'z': larger values correspond to positions higher up
 • 'color': indicates block color (optional)
 • 'visible=false': the block exists but is not seen from this view
Blocks will not float, and do not mention coordinate values or field names. Convert everything into spatial phrases (leftmost, back row, top layer, stacked above, hidden behind and so on).

2) **Ordered list format**
Views provide sequences like 'from-left-to-right' or 'from-back-to-front'. Treat them as your scan order in that view and describe the relative layout accordingly.

### ## Reasoning constraints
 • First examine the front view to infer left–right and vertical relations.
 • Then examine the left view to resolve front–back structure and occlusion.
 • Next examine the top view to determine the overall layout and coverage.
 • Finally integrate all observations into a single coherent 3D mental model.
 • When counting blocks, determine the quantity based on vertical stacking and use explicit numerical reasoning rather than purely verbal description.

### ## Output rules
 • Output only the reasoning as plain paragraphs (no headings, no lists).
 • First-person present tense ("I see... I turn... I check...").
 • Forbidden phrases: "JSON", "data", "input", "provided", "coordinates", or any literal field names like x, y, z.
 • This answer should be derived through reasoning, do not reveal that an answer was provided.

### ## Input
Problem: {PROBLEM}
(Internal check only) Answer: {ANSWER}
Three views: {THREE_VIEW_JSON}

*Figure 9.* Detailed prompt for LLM-based three-view to natural language reasoning conversion.

## C. In-Context Learning Experiment Instruction

**Instruction: Three-View Spatial Reasoning**
You are an expert visual-spatial reasoning model. You must strictly follow these steps to analyze the input image and answer the question: 1. Front View Analysis: Describe the shapes and arrangement visible when looking at the object(s) directly from the front. 2. Left View Analysis: Describe the shapes and arrangement visible when looking at the object(s) directly from the left side. 3. Top View Analysis (Plan View): Describe the shapes and arrangement visible when looking down at the object(s) from above. 4. Synthesized Reasoning: Combine the analyses from the three views to accurately locate, identify, and count the objects. 5. Final Answer: The final response must be placed inside the \boxed{} format.

## C.1 Teaching Demonstration for Block Counting

**Teaching Demonstration**
Question: How many blocks are there in the picture? Return your final response within \boxed{}.
Reasoning: Step 1: Front View Analysis I see three columns of blocks from the front. The leftmost column has no visible blocks. The middle column has two blocks stacked vertically. The rightmost column has one block at the bottom level.
Step 2: Left View Analysis Looking from the left side, I see three columns as well. The column nearest to me has two stacked blocks. The middle column has one block. The farthest column has one block.
Step 3: Top View Analysis From above, the layout is clear. I see blocks arranged in a grid of three columns (left, middle, right) and three rows (front, middle, back). The top view shows: in the middle column, there are three blocks in a vertical line from front to back; in the right column, there are two blocks — one in the middle row and one in the back row; the left column appears empty from the top.
Step 4: Synthesized Reasoning Combining the views, I can reconstruct the 3D arrangement. The front view's middle column (two blocks) corresponds to the top view's middle column having three blocks front-to-back, meaning one of them is hidden behind the front stack from the front perspective. The front view's right column (one block) matches the top view showing two blocks in that column, meaning the back one is hidden. The left view helps confirm the left column has blocks only in the back rows, not visible from the front. Counting all unique block positions from the combined grid, there are blocks at these coordinates: (middle column, front row, bottom), (middle column, middle row, bottom), (middle column, back row, bottom), (right column, middle row, bottom), (right column, back row, bottom), and (left column, back row, bottom) — six blocks in total.
Step 5: Final Answer \boxed{6}

*Figure 10.* In-Context Learning Experiment Instruction and Teaching Demonstration for Block Counting Task.

### C.2 Teaching Demonstration for Object Counting

**Teaching Demonstration**

Question: How many cylinder are there in this image? Return your final response within \boxed{}.

Reasoning: Step 1: Front View Analysis I am scanning the arrangement of 3D geometric shapes from the front perspective to establish their lateral order. Looking from left to right, I distinctly identify a sequence of objects: first a large cube, then a cone, followed by a square pyramid. Continuing the scan to the right, I see a small sphere, a larger sphere, a small cube, and a rectangular cuboid. Finally, positioned at the very far right of this lineup, I observe a single cylinder.

Step 2: Left View Analysis Shifting my perspective to the left side to assess the depth and relative positioning, I track the objects again. In this projection, I can distinguish the large cube, the cuboid, and the small sphere. I also clearly see the cylinder positioned within the depth of the field, located near the square pyramid and the cone. This view confirms the existence of the cylindrical shape observed in the front view.

Step 3: Top View Analysis Observing the scene from a top-down "bird's-eye" view allows me to verify the total inventory of all items without occlusion. From this vantage point, I can categorize every shape present. I observe a total set consisting of two spheres, two cubes, one cone, one cuboid, and one square pyramid. Crucially, in this complete collection, I identify exactly one object with the circular cross-section and vertical sides characteristic of a cylinder.

Step 4: Synthesized Reasoning By cross-referencing all three views, I can establish a definitive count. The front view highlighted one cylinder at the end of the row. The side view corroborated its position in space. The top view provided a comprehensive inventory, explicitly isolating the distinct shapes and confirming that only one of them is a cylinder. There are no duplicates or hidden cylinders.

Step 5: Final Answer \boxed{1}

### C.3 Teaching Demonstration for Object Position Reasoning

**Teaching Demonstration**

Question: In which direction is the square pyramid relative to the smaller sphere? Answer the question from the choices: ["front", "back", "right", "left", "front left", "front right", "back left", "back right"] Return your final response within \boxed{}.

Reasoning: Step 1: Front View Analysis I start by examining the front view to determine the horizontal (left-right) alignment of the objects. Looking across the scene from left to right, I see a sequence of objects: a bigger cube, a cone, a square pyramid, followed by a smaller sphere, and then others. This observation clearly establishes that the square pyramid is positioned to the left of the smaller sphere.

Step 2: Left View Analysis Next, I look at the left side view to understand the depth (back-front) relationship. In this perspective, the arrangement from left to right represents the order from the back of the scene to the front. I observe the smaller sphere appearing earlier in the sequence (further to the "left" or back), followed by the square pyramid which appears later (further to the "right" or front). This indicates that the square pyramid is located in front of the smaller sphere.

Step 3: Top View Analysis Finally, I examine the top view to confirm these spatial coordinates. From this overhead perspective, I can verify both axes simultaneously. On the left-right axis, the square pyramid is clearly to the left of the smaller sphere. On the top-down axis (representing back-to-front), the smaller sphere is positioned closer to the "top" (back) edge, while the square pyramid is closer to the "bottom" (front) edge.

Step 4: Synthesized Reasoning By synthesizing the observations from all three views, I can pinpoint the relative location: 1. Left-Right: The square pyramid is to the left of the smaller sphere. 2. Front-Back: The square pyramid is in front of the smaller sphere. Combining these two directional components, the square pyramid is located to the front left of the smaller sphere.

Step 5: Final Answer \boxed{front left}

*Figure 11.* In-Context Learning Experiment Instruction and Teaching Demonstration for Object Reasoning.

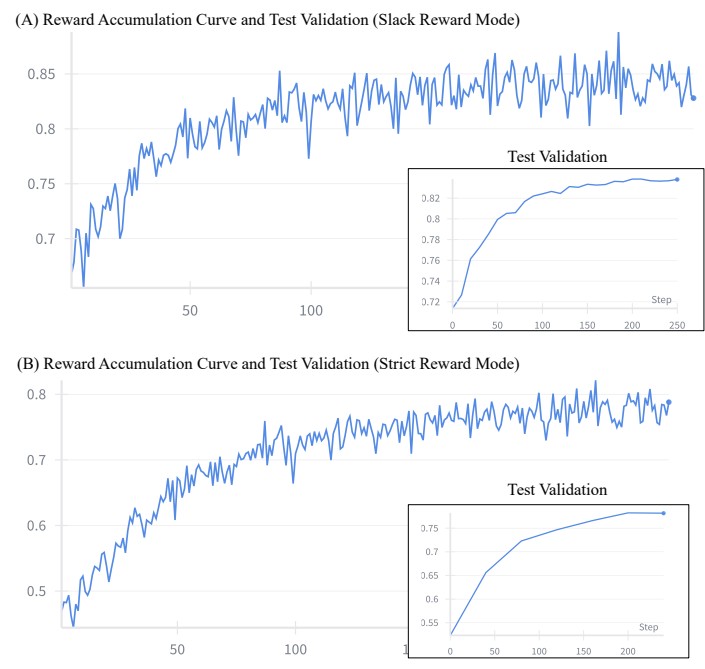

*Figure 12.* Cumulative reward curves during GRPO-based RL refinement. We report the training dynamics under two reward settings: strict reward (exact match) and slack reward (partial credit) as defined in Section 3.3.

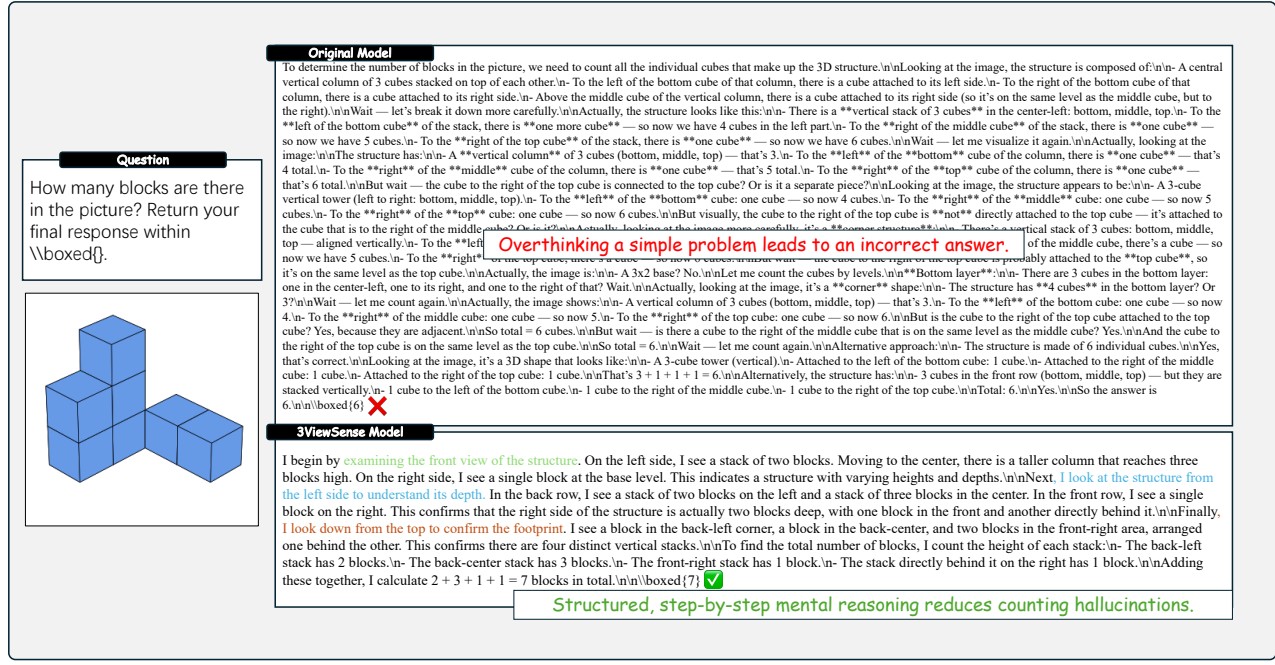

*Figure 13.* Qualitative examples of model responses on block counting under occlusion. Compared to the base model, 3ViewSense produces more concise and structured reasoning, improving robustness and accuracy.

## Example from ViewSpatial

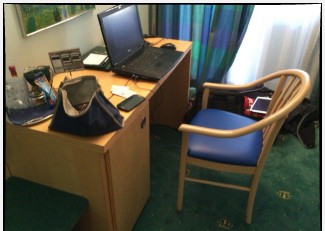

**Question:** Where is the chair in relation to the desk?Return your final response within \\boxed{}. choices: A. left\nB. above\nC. right\nD. back-up

I look at the scene from the front and see a blue chair positioned to the right of a wooden desk. The desk holds a laptop, a blue bag, and several other items. Shifting my perspective to the left side, I see the desk occupies the front-left area, while the blue chair is clearly situated further back and to the right. Looking down from above, I confirm this layout: the desk is in the front-left corner of the space, and the blue chair is placed in the back-right area. This confirms that the chair is located behind and to the right of the desk.\n\n\\boxed{C}

## Example from SPBench-SI

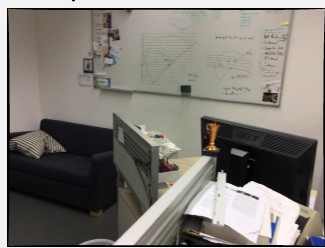

**Question:** From the camera's perspective, is the pillow to the whiteboard's left, right, front, or back? Return your final response within \\boxed{}. choices: A. left, B. right, C. front, D. back

I see a scene with a black couch on the left side, positioned toward the back of the room. Moving to the right, I encounter a whiteboard that dominates the middle and back area. Looking at the front view, the black couch is situated to the left of the whiteboard. Shifting to the left view, I observe the whiteboard at the back of the scene, while the black couch is positioned further forward. Looking down from the top view, the layout confirms that the black couch is in the leftmost column and the whiteboard is in the rightmost column, with the whiteboard being further back than the couch. Integrating these perspectives, I determine that the black couch is located to the left and in front of the whiteboard.\n\n\\boxed{A}

*Figure 14.* Case study of 3ViewSense reasoning on external benchmarks. We show qualitative examples from ViewSpatial and SPBench-SI, illustrating how 3ViewSense transfers the learned view-grounded reasoning process to external spatial reasoning tasks.

