# OpenReview forum: "3ViewSense: Spatial and Mental Perspective Reasoning from Orthographic Views in Vision-Language Models"
_ICML.cc/2026/Conference — ICML 2026 spotlight_

### Official Review · Reviewer_xgzL · 2026-03-10

**Soundness:** 3
**Presentation:** 3
**Significance:** 3
**Originality:** 3
**Overall Recommendation:** 4
**Confidence:** 4

**Summary:**

This paper addresses the limitation of current multimodal large language models (MLLMs) in lacking a coherent 3D mental representation mechanism. The authors propose 3ViewSense, a framework that introduces orthographic three-view representations (top, front, and left) as an intermediate spatial representation to facilitate spatial reasoning. The paper also presents a corresponding dataset, OrthoMind-3D, for training and evaluation. Experimental results show that the proposed method achieves significant improvements over the base model on spatial reasoning tasks, both in in-domain and out-of-domain settings.

**Compliance With Llm Reviewing Policy:**

Affirmed.

**Final Justification:**

I’m glad to see that the authors provide concrete evidence demonstrating the effectiveness of their framework, and that it truly works—particularly in terms of its strong OOD generalization performance. This is exactly what I was hoping to see, and it fully resolves my concerns. In my view, this experiment is essential for substantiating the superiority of the framework. Given that the authors have thoroughly addressed my concerns, I am inclined to accept this paper.

**Key Questions For Authors:**

Q1.
Related to Weakness 1: Could the authors provide additional evidence supporting the claim that the performance improvements are not solely due to the effectiveness of the training data? In particular, it would be helpful to better isolate the contribution of the proposed framework.

Q2.
Related to Weakness 2: Could the authors provide further analysis of the results in Table 4 and clarify why they appear inconsistent with the earlier conclusion that the primary bottleneck lies in the absence of a stable intermediate spatial representation?

I would be inclined to accept this paper if these concerns are adequately addressed.

**Limitations:**

yes

**Strengths And Weaknesses:**

**Strengths**

1. Novelty. The idea of introducing orthographic three-view representations (top, front, and left) as an intermediate spatial representation is innovative and also conceptually natural for spatial reasoning tasks. With this training framework, the model achieves significant performance improvements compared to the base model.

2. Dataset contribution. The paper also introduces a new dataset, OrthoMind-3D, designed to support training and evaluation of spatial reasoning with explicit orthographic representations.

**Weaknesses**

1. Missing ablation study. It is unclear whether the performance improvement mainly comes from the proposed 3ViewSense framework or simply from training on the newly constructed dataset. In other words, the gains may primarily stem from the effectiveness of the training data rather than the intermediate representation itself. To clarify this, it would be helpful to include an ablation where Qwen3-VL-4B-Instruct is directly fine-tuned on OrthoMind-3D using standard VQA supervision (image + question → answer), without the OMS stage or intermediate three-view generation. This would help isolate the contribution of the proposed framework.

2. Potentially inconsistent conclusions. In the in-depth analysis section, the authors argue that the primary bottleneck lies in the absence of a stable intermediate spatial representation rather than the reasoning step. However, the results in Table 4 suggest that the largest improvement mainly comes from the view-grounded reasoning SFT, while the contribution of the OMS stage appears relatively limited. This seems somewhat inconsistent with the paper’s main conclusion and would benefit from further clarification.

3. Limited task coverage. As also acknowledged in the limitation section, the proposed approach is mainly applicable to a specific category of spatial reasoning tasks. When evaluating the model on other spatial benchmarks, the authors also restrict the evaluation to selected subsets or question types. This relatively narrow coverage of spatial question types may limit the applicability of the method to more general spatial QA scenarios.

---

> ### Author Rebuttal · Authors · 2026-03-31
>
> We sincerely thank Reviewer xgzL for the constructive review and for the recognition of our novelty and dataset contribution. We particularly appreciate the explicit openness to accepting the paper if the concerns are addressed. We hope the following analysis and new evidence are convincing.
>
> **Re W1/Q1: Isolating the contribution of the framework vs. the training data.**
>
> This is the central question, and we provide three converging lines of evidence:
>
> *Evidence 1: Ablation in Table 4.*  The "VGR-SFT only" setting skips the OMS stage and directly trains the VQA capability. It achieves strong ID performance (70.3%) but weaker OOD generalization (46.6 vs. 48.5) and lower external benchmark scores (SPBench-SI: 50.2 vs. 52.6, ViewSpatial: 68.8 vs. 71.9) compared to the full OMS->VGR pipeline. This shows that OMS pre-training before VQA provides a more robust foundation, particularly under distribution shift.
>
> *Evidence 2: In-Context Learning (Figure 4).*  When we supply the same three-view spatial information to models via ICL prompts (no parameter updates), most models degrade rather than improve. This is a critical finding: if the gains were purely from data exposure, providing the same information at inference time should help. The fact that it often hurts proves the model must internalize a structured spatial thinking process through training, not merely receive spatial information.
>
> *Evidence 3: Explicit Three-View Descriptions (Table 10, Appendix C.4).*  When ground-truth three-view descriptions are provided at inference time, most models improve substantially (Table 10). Yet 3ViewSense achieves competitive performance by generating these descriptions internally, demonstrating that the framework successfully teaches the model to construct its own spatial interface.
>
> Together: (a) VGR-SFT only is helpful but insufficient for robust OOD generalization; (b) ICL with the same information can even hurt without internalization; (c) the framework's value lies in teaching the model to mentally simulate orthographic views as a reasoning scaffold.
>
> **Re W2/Q2: Apparent inconsistency in Table 4.**
>
> We appreciate this careful reading. We believe the results are fully consistent with our thesis once the role of each stage is clarified:
>
> (1) OMS-SFT alone (48.7% ID): The OMS stage trains the model to generate three-view descriptions from images, which is a fundamentally different task from answering spatial questions. Its low VQA score is expected, though still above the base model, as it was never trained to answer queries.
>
> (2) VGR-SFT only (70.3% ID, 46.6% OOD): The VGR stage is a standard VQA training process (image + question -> reasoning + answer), directly aligned with the evaluation protocol. The large improvement is natural because it explicitly trains the target capability.
>
> (3) OMS->VGR (71.0% ID, 48.5% OOD): Adding OMS before VGR yields further gains, most pronounced in OOD and external benchmarks (details below). This shows that OMS pre-training provides a stronger spatial perception foundation that benefits the subsequent VQA learning, especially under distribution shift.
>
> The value of OMS is reflected in two aspects. First, it strengthens generalization: with OMS pre-training, OOD accuracy improves from 46.6 to 48.5, SPBench-SI from 50.2 to 52.6, and ViewSpatial from 68.8 to 71.9, showing consistent gains across all OOD and external settings. Second, OMS equips the model with an explicit spatial perception capability before VQA training begins, meaning the model enters the VGR stage already capable of internally constructing three-view representations. This structured spatial grounding provides a more consistent starting point for the reasoning and RL stages, as the model no longer needs to implicitly learn spatial perception and question-answering simultaneously within a single training phase.
>
> **Re W3: Limited task coverage.**
>
> As shown in Table 2 of our paper, we have evaluated on 6 external real-world benchmarks with consistent improvements: SPBench-SI (+100%), ViewSpatial (+118%), CV-Bench 2D (+23.7%), OmniSpatial (+2.9%). We additionally evaluated on MindCube, a cognitive-oriented benchmark that tests spatial mental modeling under occlusion:
>
> | Model             | MindCube-tiny | MindCube |
> | - | - | - |
> | Qwen3-VL-4B (base)   | 23.2    | 21.6   |
> | 3ViewSense-4B (Ours) | 38.2    | 33.1   |
>
> These results demonstrate that 3ViewSense generalizes effectively to diverse real-world scenarios beyond our training domain. Notably, our training data contains only block-counting and simple object reasoning, yet the model improves on substantially different tasks including camera perspective understanding, egocentric reasoning, and mental modeling, suggesting the learned ability transfers as a general spatial perception skill. We hope to explore broader task coverage in future work.
>
> We hope these analyses adequately address the concerns, and we sincerely thank the reviewer for the constructive feedback.

---

> > ### Author Rebuttal · Reviewer_xgzL · 2026-04-03
> >
> > The authors have satisfactorily addressed my concerns regarding the apparent inconsistency in the conclusions and the issue of limited task coverage. However, my main concern about disentangling the contribution of the proposed 3ViewSense framework from that of the training data remains insufficiently addressed.
> >
> > Specifically, I find the provided evidence unconvincing:
> >
> > (1) Regarding Evidence 1 (Table 4):
> > The reported improvements are relatively small (less than 3% across benchmarks), which may fall within the range of variance. More importantly, this experiment mainly shows that each component within the proposed framework ncontributes to performance. However, this is expected, as training the full pipeline versus only part of it typically leads to improvements.
> > Crucially, this comparison does not address the core question raised in my original review: whether the gains come from the framework itself or simply from the training data. To properly isolate the contribution of the framework, it is necessary to compare against a standard QA training paradigm using the same data (i.e., directly fine-tuning the base model with image + question → answer supervision on OrthoMind-3D). Without such a controlled comparison, the effectiveness of the proposed framework remains unclear.
> >
> > (2) Regarding Evidence 2 and 3:
> >  Neither piece of evidence directly addresses the core concern. Evidence 2 (ICL) mainly shows that models cannot utilize the provided spatial information without appropriate training, which reflects a lack of internalization rather than demonstrating the advantage of the proposed framework over standard training. Evidence 3 introduces additional privileged information at inference time (ground-truth three-view descriptions), and the resulting improvement is therefore expected.
> > Overall, these results neither isolate nor quantify the contribution of the framework itself, and are thus not directly relevant to distinguishing framework effectiveness from data effects.
> >
> > In summary, the current rebuttal still does not adequately resolve the key question of whether the observed gains stem from the proposed framework or from the newly constructed dataset. A controlled ablation with standard VQA training on the same data would be necessary to convincingly support the claims.
> >
> > Given that my primary concern remains insufficiently addressed, I will maintain my original score.

---

> > > ### Author Response · Authors · 2026-04-03
> > >
> > > We sincerely thank Reviewer xgzL for the continued patience and the clear feedback. Upon re-reading the reply, we realized that we had misunderstood the core question: the reviewer was not asking whether each stage within our pipeline contributes, but whether the framework itself outperforms a standard QA paradigm on the same data. We sincerely apologize for this oversight. As soon as we recognized this, we immediately reorganized the training data and launched the requested ablation.
> > >
> > > **Direct QA Ablation.**
> > >
> > > Following the reviewer's description, we fine-tuned Qwen3-VL-4B-Instruct directly on OrthoMind-3D using standard VQA supervision (image + question → answer), without the OMS stage or intermediate three-view generation. All hyperparameters are kept identical to VGR-SFT. We note that Direct QA fine-tuning tends to induce catastrophic forgetting of the base model's instruction-following behavior, which made answer extraction during evaluation somewhat more involved. We took additional care in answer parsing and matching to ensure a fair comparison.
> > >
> > > Results:
> > >
> > > | Model              | Method                       | ID   | OOD  | SPBench-SI | ViewSpatial |
> > > | -------------------- | ------------------------------ | ------ | ------ | ------------ | ------------- |
> > > | Base (Qwen3-VL-4B) | -                            | 45.0 | 44.2 | 27.1       | 33.5        |
> > > | Direct QA          | SFT: image+Q → answer       | **80.3**     | **49.8**     | 16.7       | 28.6        |
> > > | VGR-SFT-Only       | SFT: 3ViewSense CoT + answer | 70.3 | 46.6 | **50.2**           | **68.8**            |
> > >
> > > We believe this directly addresses the reviewer's core concern:
> > >
> > > (1) Direct QA achieves higher in-domain accuracy, which is expected: without intermediate steps, the model memorizes input-output mappings efficiently and performs well on data resembling the training distribution.
> > >
> > > (2) However, on external benchmarks, Direct QA falls below the untrained base model on both SPBench-SI and ViewSpatial. The data alone, when used in a standard QA setting, causes overfitting and erodes the model's pre-existing spatial capabilities.
> > >
> > > (3) VGR-SFT, trained on the same data with three-view CoT, improves consistently on external benchmarks. The same data produces quite different outcomes depending on whether intermediate spatial reasoning is involved.
> > >
> > > To connect back to the reviewer's precise framing: this comparison isolates and quantifies the contribution of the framework itself, independent of the data. The answer is that the data under standard QA training is actively harmful to generalization, while the three-view reasoning process transforms the same data into a source of transferable spatial understanding.
> > >
> > > Additionally, to provide more concrete evidence of generalization, we have prepared qualitative case studies showing how 3ViewSense reasons on external benchmarks (ViewSpatial, SPBench-SI, and MindCube) with question types that differ considerably from those in OrthoMind-3D. The model's detailed outputs can be viewed through this anonymous link (case study): https://anonymous.4open.science/r/Image_Anonymous-4326. Across these diverse tasks, the model consistently applies its learned multi-view decomposition strategy, further illustrating that the framework teaches a transferable reasoning approach rather than task-specific patterns.
> > >
> > > We are deeply grateful for the reviewer's patience, rigor, and willingness to engage in multiple rounds of detailed discussion. This exchange has improved our work. We will incorporate the Direct QA ablation results and these case examples into the final version of the paper. We sincerely hope these results resolve the remaining concern, and we would be truly grateful if the reviewer could consider re-evaluating our work.

---

### Official Review · Reviewer_uhgW · 2026-03-13

**Soundness:** 3
**Presentation:** 3
**Significance:** 3
**Originality:** 3
**Overall Recommendation:** 5
**Confidence:** 3

**Summary:**

This paper highlights a gap in VLMs: they excel at symbolic logic but struggle with basic spatial tasks like counting stacked blocks under occlusion. The authors find this is due to the lack of a view-consistent intermediate representation that links egocentric perception and reasoning. To fix this, they introduce 3ViewSense, a Simulate-and-Reason framework inspired by orthographic projections. It has two stages: (1) Orthographic Mental Simulation (OMS), training the model to produce structured view descriptions from a single egocentric image, and (2) View-Grounded Reasoning (VGR), training the model to solve spatial queries based on these views. They also present OrthoMind-3D, a benchmark for block counting and object reasoning tasks. Results show improvements on in-domain and out-of-domain spatial tasks, with GRPO-based RL refinement enhancing performance.

**Compliance With Llm Reviewing Policy:**

Affirmed.

**Final Justification:**

The paper presents a well-motivated framework for spatial reasoning in VLMs, grounded in a clean diagnostic finding that the bottleneck is a missing view-consistent intermediate representation rather than insufficient perception or reasoning. The two-stage Simulate-and-Reason design is principled, and the OrthoMind-3D benchmark with its bijective uniqueness condition is carefully constructed. Results are strong both in-domain and on transfer benchmarks such as SPBench-SI and ViewSpatial. The rebuttal strengthened the paper by providing 8B scaling results, clarifying reward design tradeoffs, and explaining failure modes for object positioning. The framework is inherently scoped to settings where orthographic projections are informative, and the authors acknowledge this limitation openly. While generalization to complex real-world scenes with irregular shapes remains future work, the current contributions are solid and the approach offers a scalable path toward stronger spatial intelligence. I raise my score to 5.

**Key Questions For Authors:**

1. How does the approach perform with larger base models?

2. OMS stage needs orthographic annotations for supervision. For real-world deployment beyond synthetic environments, how would you get these annotations? Have you considered self-supervised or weakly-supervised methods for learning mental simulation?

3. In table 2, the strict vs. slack reward comparison shows trade-offs. When is each preferable? Slack reward seems better for OOD generalization. Does strict reward cause overfitting?

4. The paper show significant improvements in block counting, but performs relatively worse on object reasoning, especially object position. What are the main failure modes for object positioning, and do they relate to orthographic representation limitations?

5. If the OMS stage produces incorrect views, how robust is the VGR stage to these errors?

**Limitations:**

yes

**Strengths And Weaknesses:**

### Strengths

S1: The paper suggests the spatial intelligence gap in VLMs is an interface issue, not perception or reasoning, due to lack of structured spatial representation. Diagnostic experiments support this.

S2: The Simulate-and-Reason framework, inspired by engineering, uses a principled OMS and VGR stages. Ablation shows VGR-SFT alone is insufficient without OMS pre-training.

S3: The results are impressive, especially the dramatic improvements on spatial reasoning benchmarks. The model nearly perfects in-domain tasks and transfers well out-of-domain. The RL refinement analysis in Fig. 5 revealing stable training from VGR-SFT but instability from OMS-SFT is insightful.

S4: The OrthoMind-3D benchmark is a well-designed diagnostic tool with a bijective mapping between 3D configurations and 2D projections. Its rigorous data construction, including in-domain and out-of-domain splits, reflects good practice.

S5: The analysis shows the base model's verbose outputs versus 3ViewSense's concise answers, highlighting the benefit of structured intermediate representations.

### Weaknesses

W1: The framework is designed for block-world and simple object reasoning where orthographic projections work well. Its ability to handle complex real-world tasks with irregular shapes, occlusions, or where orthographic views fall short is unclear. The paper does not address this limitation.

W2: Relying synthetic training data and AI-generated images with manual verification raises scalability issues. Generating OMS supervision needs 3D structure for orthographic annotations, creating a chicken-and-egg problem for real-world images.

W3: The base model is small, and the paper doesn't explore if larger VLMs have better spatial intelligence or if the framework scales up. This matters because the spatial intelligence gap may close with scale.

W4: The paper suggests the approach is inspired by engineering drawings that define 3D structures through standard projections, but this is mainly metaphorical. Real drawings include precise dimensions, tolerances, and annotations, which go beyond the structured text descriptions used here.

---

> ### Author Rebuttal · Authors · 2026-03-31
>
> We sincerely thank Reviewer uhgW for the thorough and insightful review, and for the recognition of our diagnostic analysis, framework design, and empirical results. We address each question and weakness below.
>
> **Re Q1 & W3: Performance with larger base models?**
>
> Figure 4 (B) provides indirect evidence that even the most capable proprietary models (GPT-5, Gemini-3-pro) benefit dramatically from three-view descriptions, confirming that the spatial interface gap persists across scales.
>
> To better address the concerns of the reviewers, we have also trained an 8B variant. ID/OOD results (accuracy %):
>
> | Model         | Size | ID-BlkC | ID-ObjC | ID-ObjP | OOD-BlkC | OOD-ObjC | OOD-ObjP |
> | - | - | - | - | - | - | - | - |
> | 3ViewSense-4B | 4B   | 94.4    | 98.7    | 92.3    | 38.7     | 50.9     | 76.1     |
> | 3ViewSense-8B | 8B   | 96.6    | 98.3    | 92.6    | 39.3     | 62.0     | 79.8     |
>
> The 8B model consistently matches or surpasses the 4B variant, with the most notable gains on OOD, suggesting that larger models benefit even more from our framework in terms of generalization.
>
> **Re Q2 & W2: OMS annotations for real-world deployment?**
>
> We appreciate this question. Notably, our current approach already demonstrates strong sim-to-real transfer: the model is trained exclusively on synthetic data with programmatically generated OMS annotations, yet transfers effectively to real-world benchmarks with entirely different visual styles, including SPBench-SI, ViewSpatial, and MindCube. This suggests that the spatial reasoning ability learned from synthetic OMS supervision generalizes well beyond the synthetic domain. For future work, additional paths could further improve scalability, such as using strong VLMs as teachers to generate pseudo-annotations or exploring weakly-supervised approaches where only the final answer signal trains OMS through RL.
>
> **Re Q3: Strict vs. slack reward trade-offs?**
>
> From Table 2, strict reward tends to achieve slightly higher ID accuracy, while slack reward yields better OOD generalization and external benchmark transfer. Strict reward concentrates gradient on exact-match trajectories, effective in-domain but offering no signal for near-correct outputs. Slack reward provides denser feedback from partially correct reasoning, promoting more robust generalization.
>
> **Re Q4: Object position failure modes?**
>
> In our Object Reasoning task (Section 3.2), spatial relations are discretized into 8 directions, and we apply a 5-degree tolerance margin near canonical axes to handle boundary ambiguity. The main failure mode occurs when an object's relative direction falls near the boundary between two adjacent sectors (e.g., between "front" and "front-left"), where even small estimation errors can lead to an incorrect discrete label. Additionally, the OMS representation uses ordered perceptual sequences (e.g., left-to-right scan) from each canonical view, which is naturally suited for counting and ordering but less informative for precise angular discrimination.
>
> **Re Q5: VGR robustness to OMS errors?**
>
> We would like to first note that the OMS training annotations are programmatically generated from the 3D scene structure, so the supervision signal itself contains no errors. The concern thus applies to inference time, where the model's self-generated OMS outputs may be imperfect.
>
> Our GRPO-based RL stage addresses this naturally: since OMS and VGR are jointly rolled out during RL, if an inaccurate output leads to a wrong final answer, the low reward signal penalizes the entire trajectory end-to-end. This incentivizes the model to produce outputs that are maximally useful for correct downstream reasoning. As shown in Figure 5, the steadily increasing reward curve from the VGR-SFT initialization confirms that this end-to-end optimization is effective and stable.
>
> **Re W1: Generalization.**
>
> Our cross-benchmark transfer results in Table 2 provide strong evidence: SPBench-SI (27.1->54.2, +100%), ViewSpatial (33.5->72.9, +118%), CV-Bench 2D (68.8->85.1, +23.7%) and OmniSpatial (61.8->67.6, +2.9%). These benchmarks involve diverse natural images and spatial layouts well beyond block-world structures. As discussed in Appendix D, not all spatial reasoning problems can be fully addressed by three orthographic views alone, as some tasks require additional physical and semantic priors beyond geometry. We consider extending to such scenarios a valuable future direction.
>
> **Re W4: Engineering drawing analogy.**
>
> We agree that our method is inspired by, rather than replicating, engineering drawings. The core principle we borrow is "using standardized projections to eliminate geometric ambiguity," not precise dimensioning or tolerancing. This parallels how Chain-of-Thought is inspired by human reasoning without replicating the full cognitive process.
>
> We hope these responses address all concerns. We are grateful for the careful reading and constructive questions.

---

> > ### Author Rebuttal · Reviewer_uhgW · 2026-04-04
> >
> > Thank you for the detailed rebuttal. The 8B variant results showing consistent gains, especially on OOD generalization, address my concern about scalability. The sim-to-real transfer evidence across SPBench-SI, ViewSpatial, and MindCube is convincing for the synthetic data concern. The analysis of strict vs slack reward tradeoffs and the object positioning failure modes near sector boundaries are clear and informative. The explanation that GRPO naturally handles OMS errors through end-to-end reward signals is well-supported by the stable reward curves in Figure 5. My concerns are adequately addressed.

---

### Official Review · Reviewer_WwTp · 2026-03-13

**Soundness:** 3
**Presentation:** 4
**Significance:** 2
**Originality:** 2
**Overall Recommendation:** 5
**Confidence:** 2

**Summary:**

The manuscript strives to explore a central concept: whether explicitly inducing orthographic projections can provide a stable intermediate representation for spatial reasoning in vision-language models. The authors appear to study a core issue in multimodal reasoning — the difficulty of constructing coherent 3D mental representations from 2D observations — and propose a two-stage simulate-and-reason framework that generates orthographic views and then performs reasoning over them. Experiments on the proposed OrthoMind-3D dataset and several spatial benchmarks show substantial improvements over existing open-source models. However, the method introduces strong task-specific inductive bias and relies on structured intermediate representations whose generality to more complex real-world spatial reasoning tasks remains unclear.

**Compliance With Llm Reviewing Policy:**

Affirmed.

**Final Justification:**

The paper is overall well written and the method is sound. Generalization evidences presented in rebuttal seems promising.

**Key Questions For Authors:**

- The orthographic views are encoded using a structured JSON-style format. Did the authors experiment with alternative formats (e.g., natural language descriptions, more compact symbolic encodings, or different ordering)? How sensitive is performance to this representation design?
- Many tasks in OrthoMind-3D involve block structures or objects placed on simple planes. How does the method perform on scenes with irregular shapes, continuous geometry, or more complex spatial arrangements?

**Limitations:**

yes (in appendix D)

**Strengths And Weaknesses:**

# Strengths

***Clear motivation and intuitive method design***: The manuscript strives to explore a central concept: addressing the “spatial intelligence gap” in vision-language models by introducing an explicit spatial interface for reasoning. The authors appear to study a core issue in multimodal reasoning—how models convert 2D observations into coherent spatial representations—and propose a structured simulate-and-reason pipeline based on orthographic projections. The overall motivation is clear and the conceptual framing is easy to follow.

***Strong presentation with helpful visualizations***: The paper is well presented and includes numerous figures that help build intuition about the method and its reasoning process. In particular, the diagrams illustrating the orthographic-view representation and the two-stage pipeline make the approach easy to understand. These visual explanations significantly improve the accessibility of the method.

***Strong empirical improvements***: The reported gains on the proposed OrthoMind-3D benchmark and several external spatial reasoning benchmarks are substantial, especially for block-counting and occlusion-heavy reasoning tasks. These results suggest that introducing structured spatial representations can meaningfully improve the reasoning behavior of current VLMs.

---

# Weaknesses

***Limited evidence of generalization beyond structured spatial tasks***: The proposed representation introduces strong inductive bias toward grid-like spatial structures that can be easily described using orthographic projections. Many evaluation tasks involve block counting or objects placed on simple planes, which are naturally suited to this representation. It remains unclear how well the approach would generalize to more complex real-world spatial reasoning tasks involving irregular objects, continuous geometry, or more diverse scene layouts.

***Unclear role of representation design vs. reasoning improvement***: The method relies on generating a structured orthographic-view representation before reasoning. However, it is not entirely clear whether the improvements arise from better spatial reasoning capabilities or from providing the model with a carefully engineered intermediate representation that simplifies the reasoning problem.

***Sensitivity to representation formatting is not studied***: In the examples (e.g., Figure 7), the orthographic views are represented using verbose structured descriptions (e.g., JSON-like formats). It would be helpful to understand whether the specific formatting plays an important role in performance. The paper does not provide ablations exploring alternative representations or simpler formats, leaving open the possibility that the gains may depend on this particular encoding.

***Limited analysis of applicability to broader spatial reasoning settings***: While the orthographic-view approach is well motivated for certain spatial tasks, it is less clear whether it would scale to scenarios where spatial reasoning involves richer semantics, complex geometry, or interactions between objects. Additional experiments or discussion of such settings would strengthen the claims about addressing the spatial intelligence gap.

---

> ### Author Rebuttal · Authors · 2026-03-31
>
> We sincerely thank Reviewer WwTp for the thoughtful and encouraging review. We are particularly grateful for the recognition of our clear motivation, intuitive method design, strong presentation with helpful visualizations, and substantial empirical improvements. These positive assessments mean a great deal to us. We address each concern below.
>
> **Clarification: Our model does NOT receive external three-view inputs.**
>
> We would like to first clarify an important point: 3ViewSense does not take pre-computed orthographic views as input. Instead, from a single 2D egocentric image, the OMS stage teaches the model to internally generate structured three-view descriptions as intermediate reasoning artifacts. The VGR stage then reasons over these self-generated "mental sketches." Our contribution is teaching the model a new spatial thinking process, not providing it with extra visual data.
>
> **Re W1 & Q2: Generalization beyond structured spatial tasks.**
>
> We respectfully note that our method does not require input scenes to be grid-like. The model learns a general "mental simulation" ability, inferring orthographic projections from arbitrary 2D images. Evidence:
>
> (1) Our OOD data uses a sandbox engine with stochastically scattered blocks forming unstructured high-entropy piles, captured from diverse natural viewpoints, very different from the grid-structured ID data.
>
> (2) On completely independent third-party benchmarks (in Table 2): SPBench-SI (27.1->54.2, +100%), ViewSpatial (33.5->72.9, +118%) and so on. These involve natural images with various spatial layouts, not block-world structures.
>
> (3) We additionally evaluated on MindCube, a cognitive-oriented mental modeling benchmark: 3ViewSense improves from 23.2 to 38.2 (+15.0pp on MindCube-tiny) and from 21.6 to 33.1 (+11.5pp on MindCube), further demonstrating transfer beyond our training domain.
>
> As discussed in Appendix D, not all spatial reasoning problems can be fully addressed by three orthographic views alone, as some tasks require additional physical and semantic priors beyond geometry. We consider extending to such richer scenarios a valuable future direction.
>
> **Re W2: Role of representation design vs. reasoning improvement.**
>
> Our diagnostic analysis (Section 1, Figure 4 and Appendix C.3) demonstrates that (a) the visual encoder already extracts sufficient geometric information (probe accuracy: 55.8%), and (b) reasoning dramatically improves when a view-consistent interface is provided. These together show the bottleneck is a missing spatial interface, not insufficient perception or reasoning.
>
> Crucially, because our model generates the three-view representations internally from a single image, the representation capability is itself a form of reasoning improvement: the model learns a new way of spatial thinking. This is analogous to teaching someone to draw engineering sketches, where the drawing process itself deepens spatial understanding. The representation and reasoning improvements are thus deeply intertwined rather than separable.
>
> **Re W3 & Q1: Sensitivity to representation formatting.**
>
> The JSON-style format is a practical choice for our data synthesis stage, where machine-readable structure simplifies automated generation and verification. We believe the decisive factor for performance is whether the intermediate description captures sufficient spatial information from the three canonical views, not the specific encoding. Indeed, Appendix Figure 7 shows that the same spatial content can be equivalently expressed in natural language via rule-based conversion.
>
> Our ablation in Table 4 (Section 5.3) further shows that the OMS stage itself is essential: omitting it entirely ("VGR-SFT only") leads to weaker OOD generalization (46.6 vs. 48.5) and lower transfer to external benchmarks (SPBench-SI: 50.2 vs. 52.6, ViewSpatial: 68.8 vs. 71.9). This confirms that learning to produce spatially informative intermediate representations is the key factor driving improvements.
>
> We hope these clarifications address the reviewer's concerns. We again sincerely appreciate the recognition of our work and the encouraging assessment.

---

> > ### Author Rebuttal · Reviewer_WwTp · 2026-04-04
> >
> > Answers resolve my questions. Raising my score.

---

### Official Review · Reviewer_Lv2q · 2026-03-13

**Soundness:** 3
**Presentation:** 3
**Significance:** 3
**Originality:** 3
**Overall Recommendation:** 4
**Confidence:** 4

**Summary:**

This paper addresses the "spatial intelligence gap" in Vision-Language Models (VLMs), where models with high-level logical reasoning still fail at basic spatial tasks such as block counting under occlusion. Through a diagnostic study, the work demonstrates that the bottleneck lies in the lack of a stable, view-consistent spatial interface rather than deficiencies in the visual encoder. To address this issue, the paper proposes 3ViewSense, a Simulate-and-Reason framework that decomposes 2D scenes into canonical orthographic projections (front, top, and left), enabling models to construct more reliable 3D mental representations. The authors also introduce OrthoMind-3D, a diagnostic benchmark comprising both In-Domain synthetic data and Out-of-Domain sandbox-engine data to evaluate perspective reasoning and occlusion handling.

**Compliance With Llm Reviewing Policy:**

Affirmed.

**Final Justification:**

My concerns are fully addressed and hence, I retain my positive score. Please include all additional discussions and experiments in the revised version.

**Key Questions For Authors:**

1. Why not comparing with leading spatial intelligence models? To better contextualize the results, it would be valuable to compare 3ViewSense with recent models explicitly designed for spatial reasoning tasks, such as Cambrian-S or SenseNova-SI. These models also target spatial intelligence challenges including perspective taking and global rotation (conceptually related to the three-view setting).
2. Is JSON/structured formatting critical? The current setting functions similarly to a Chain-of-Thought (CoT) process. The authors could conduct an ablation study to determine if the specific JSON/structured output format is necessary. Specifically, the authors could test a variant that removes the structured formatting in Stage I and instead using Stage 2 teacher model generated data (natural-language descriptions of the three views) or directly trains the model to produce the final answer (a direct QA setting). This would help isolate whether the performance gains stem primarily from the spatial information itself.
3. Can it generalize to real-world scenarios? While the paper mentions general benchmarks in Appendix C.1, it would be helpful to provide concrete examples for these benchmarks. It would also strengthen the paper to evaluate the framework on subsets of VSI-Bench or MindCube. Since these benchmarks provide multiple global angle images of the same scene (functionally similar to the information in the three-view JSON), showing whether the "Simulate-and-Reason" mechanism offers comparable gains there would significantly strengthen the claims regarding real-world applicability.

**Limitations:**

Yes.

**Strengths And Weaknesses:**

**Strengths:**
+ The diagonistic analysis is insightful. The paper provides a compelling investigation into the visual encoder's capabilities. By freezing visual features and training a lightweight probe that achieves 55.8% accuracy, the authors effectively argue that the encoder already extracts useful geometric information, but the downstream reasoning pipeline fails to fully utilize it.
+ The effectiveness has been validated on leading proprietary models. The methodology demonstrates clear utility for closed-source models. Providing 3-view hints to models like GPT-5 and Gemini-3-pro results in significant performance gains.
+ The framework is systematically established. The separation of the task into Orthographic Mental Simulation (OMS) and View-Grounded Reasoning (VGR) stages, supported by GRPO-based reinforcement learning, provides a transparent and reproducible pipeline for improving spatial grounding.

**Weaknesses:**
- A potential limitation is the similarity between the In-Domain (ID) and Out-of-Domain (OOD) datasets. While the sandbox engine introduces stochastic variations, the voxel-based nature of both sets may not sufficiently challenge the model’s generalization to truly unstructured, real-world environments compared to more diverse spatial benchmarks.

---

> ### Author Rebuttal · Authors · 2026-03-31
>
> We sincerely thank Reviewer Lv2q for the thoughtful review and for recognizing our diagnostic insight, empirical validation, and systematic framework design. We address each concern below.
>
> **Re W1: ID/OOD dataset similarity.**
>
> We appreciate this important observation. While both subsets involve voxel-based structures, the OOD data differs substantially: (1) it is generated by a completely different engine (sandbox game engine vs. programmatic synthesis); (2) blocks are stochastically scattered into unstructured, high-entropy piles rather than placed on regular grids; (3) images are captured from diverse natural viewpoints with perspective variations absent from ID data.
>
> More importantly, our model transfers well to entirely **independent third-party benchmarks (in Table 2)** : SPBench-SI (27.1->54.2, +100%), ViewSpatial (33.5->72.9, +118%) and so on. These benchmarks involve different visual styles, scene types, and question formats, providing strong evidence of genuine generalization beyond the voxel domain.
>
> **Re Q1: Comparison with SenseNova-SI.**
>
> We note that Cambrian-S primarily targets video understanding, which differs from our single-image setting. SenseNova-SI shares a more similar problem scope, so we have included it for comparison. Our paper already compares extensively with proprietary models (Table 1, Figure 4) and spatial-reasoning models (Table 2) at similar or larger scales.
>
> Following your suggestion, we evaluated latest SenseNova-SI variants on OrthoMind-3D. BlkC = Block Counting, ObjC = Object Counting, ObjP = Object Position; suffix "-A" denotes attribute-conditioned queries.
>
> ID results (accuracy %):
>
> | Model                      | Size | BlkC | BlkC-A | ObjC | ObjP | ObjC-A | ObjP-A |
> | ---------------------------- | ------ | ------ | -------- | ------ | ------ | -------- | -------- |
> | SI-1.3-InternVL3           | 8B   | 15.2 | 50.5   | 57.3 | 30.7 | 71.0   | 42.6   |
> | 3ViewSense-rl-slack (Ours) | 4B   | **94.4**     | **88.6**       | **98.7**     | **92.3**     | **98.4**       | **93.4**       |
>
> OOD results (accuracy %):
>
> | Model                      | Size | BlkC | BlkC-A | ObjC | ObjP |
> | ---------------------------- | ------ | ------ | -------- | ------ | ------ |
> | SI-1.3-InternVL3           | 8B   | 15.7 | 62.1   | 48.2 | 20.2 |
> | 3ViewSense-rl-slack (Ours) | 4B   | **38.7**     | **70.2**       | **50.9**     | **76.1**     |
>
> Despite using a smaller 4B backbone, 3ViewSense outperforms SenseNova-SI by large margins on both ID and OOD, demonstrating strong generalization in these tasks.
>
> **Re Q2: Is JSON/structured formatting critical?**
>
> The JSON format is adopted in the OMS stage primarily for programmatic data synthesis, as it facilitates automated generation, validation, and quality control during the data construction pipeline. We also provide a rule-based conversion from JSON format to natural language in Appendix Figure 7 for clarity. The specific format is not the decisive factor; what truly matters is that the descriptions contain sufficient and accurate spatial information from the three canonical views.
>
> Our Table 4 (Section 5.3) provides supporting evidence. The "VGR-SFT only" row trains without the OMS alignment stage, and the results show weaker OOD generalization (46.6 vs. 48.5) and lower external benchmark scores. This confirms that the key contribution of the OMS stage lies in teaching the model to produce spatially informative intermediate representations, rather than in the particular formatting choice.
>
> **Re Q3: Generalization to real-world scenarios.**
>
> Our main experiments in Table 2 already include cross-benchmark transfer results on CV-Bench, OmniSpatial, SPBench-SI, and ViewSpatial, all of which use natural images with diverse spatial layouts. As discussed in Appendix D, we acknowledge that not all spatial reasoning problems can be fully addressed by three orthographic views alone, as some tasks require additional physical and semantic priors beyond geometry (e.g., support relations, affordances, and dynamics).
>
> Regarding VSI-Bench: it is primarily a video understanding benchmark with multi-view inputs, which differs from our single-image setting. Following your suggestion on MindCube, we evaluated our model using the EASI evaluation script:
>
> | Model                         | MindCube-tiny | MindCube |
> | ------------------------------- | --------------- | ---------- |
> | Qwen3-VL-4B (base)            | 23.17         | 21.63    |
> | 3ViewSense-4B-rl-slack (Ours) | 38.17         | 33.07    |
>
> 3ViewSense achieves +15.0pp and +11.44pp improvements respectively, demonstrating that our framework's spatial reasoning capability transfers effectively to cognitive-oriented mental modeling benchmarks beyond our training domain.
>
> We hope these clarifications and new results address the reviewer's concerns. We again sincerely appreciate the recognition of our work.

---

> > ### Author Rebuttal · Reviewer_Lv2q · 2026-04-02
> >
> > Thank you for the clarifications and additional results. The responses are helpful, but I still believe both points would benefit from more precise isolation and stronger empirical support.
> >
> > **Regarding Q2 (structured output format):**
> > While Table 4 provides useful evidence on the importance of the OMS stage, it still does not directly answer whether the structured output format itself is necessary, whether the improvements primarily come from the inclusion of richer spatial information in the intermediate outputs.
> >
> > To better isolate this factor, it would be helpful to include a more direct ablation. In particular, an experiment that trains the model in a direct question-answering setting on the same data, without any intermediate structured output, would provide a clearer baseline. For example, instances such as those in Figure 12 could be reformulated into direct QA pairs (e.g., “How many blocks are there in the picture……?” → “7”) without requiring intermediate representations.
> > This would help determine whether the gains come from:
> > - the presence of explicit intermediate spatial reasoning, or
> > - simply from additional supervision data
> >
> > I would also appreciate seeing this baseline evaluated on both OrthoMind-3D and broader spatial reasoning benchmarks to better understand its generality.
> >
> > **Regarding Q3 (generalization to real-world scenarios):**
> > The additional MindCube results are valuable and strengthen the paper. To further clarify the generalization capability, it would be helpful to provide quantitative examples or case studies from general benchmarks (e.g., MindCube, ViewSpatial), particularly for question types that differ from those in the OrthoMind-3D dataset.
> >
> > Such examples would offer more concrete insight into how well the proposed approach transfers to diverse reasoning settings beyond the original training distribution.

---

> > > ### Author Response · Authors · 2026-04-03
> > >
> > > We sincerely thank Reviewer Lv2q for the patience and detailed guidance throughout this discussion. We owe the reviewer an honest acknowledgment: we initially misunderstood the core intent of Q2. We had interpreted it as asking about the relative contributions of the OMS vs. VGR stages, whereas the reviewer was in fact raising a more fundamental question, namely whether the gains stem from explicit intermediate spatial reasoning or simply from the supervision data. We sincerely apologize for this oversight, and we are deeply grateful that the reviewer took the time to clarify the exact experiment needed.
> > >
> > > Upon carefully re-reading the follow-up, we immediately recognized the gap and tried our best to conduct the requested controlled ablation.
> > >
> > > **Direct QA Ablation.**
> > >
> > > We constructed a Direct QA baseline following the reviewer's suggestion precisely: the same OrthoMind-3D training data reformulated into standard VQA pairs (image + question → answer only), with all intermediate three-view reasoning removed. The model (Qwen3-VL-4B) is fine-tuned under identical hyperparameters. We note that Direct QA fine-tuning tends to induce catastrophic forgetting of the base model's instruction-following behavior, which made answer extraction during evaluation somewhat more involved. We took additional care in answer parsing and matching to ensure a fair comparison.
> > >
> > > Results:
> > >
> > > | Model              | Method                       | ID   | OOD  | SPBench-SI | ViewSpatial |
> > > | -------------------- | ------------------------------ | ------ | ------ | ------------ | ------------- |
> > > | Base (Qwen3-VL-4B) | —                           | 45.0 | 44.2 | 27.1       | 33.5        |
> > > | Direct QA          | SFT: image+Q → answer       | **80.3**     | **49.8**     | 16.7       | 28.6        |
> > > | VGR-SFT-Only       | SFT: 3ViewSense CoT + answer | 70.3 | 46.6 | **50.2**           | **68.8**            |
> > >
> > > The results reveal a striking pattern:
> > >
> > > (1) *Direct QA achieves higher in-domain accuracy.*  This is expected: without intermediate reasoning steps, the model directly memorizes input-output mappings, yielding strong performance on data resembling the training distribution.
> > >
> > > (2) *However, Direct QA catastrophically degrades on external benchmarks*, falling even below the untrained base model on both SPBench-SI and ViewSpatial. This reveals that standard QA training on OrthoMind-3D causes severe overfitting while damaging the model's pre-existing spatial capabilities.
> > >
> > > (3) *In contrast, VGR-SFT with three-view CoT improves consistently on external benchmarks*, with clear gains on both SPBench-SI and ViewSpatial over the base model. The same data produces quite different outcomes depending on whether explicit intermediate spatial reasoning is involved.
> > >
> > > This controlled comparison directly answers the reviewer's question: the training data alone is not only insufficient but actively harmful to generalization. The gains on external benchmarks are entirely attributable to the intermediate spatial reasoning process, which teaches a transferable spatial thinking strategy rather than dataset-specific shortcuts.
> > >
> > > **Qualitative Examples from External Benchmarks.**
> > >
> > > Following the reviewer's suggestion, we provide case studies showing how 3ViewSense reasons on external benchmarks with question types absent from OrthoMind-3D. The model's detailed outputs can be viewed through this anonymous link (case study results): https://anonymous.4open.science/r/Image_Anonymous-4326
> > >
> > > Across ViewSpatial (real-world relative position), SPBench-SI (camera-centric perspective reasoning), and MindCube (multi-image cognitive spatial reasoning), 3ViewSense consistently applies its learned multi-view decomposition strategy to arrive at correct answers. These tasks are fundamentally different from our training distribution, demonstrating that the framework acquires a generalizable spatial reasoning methodology that transfers effectively to diverse real-world scenarios.
> > >
> > > We are deeply grateful for the reviewer's patience and precise guidance, which helped us identify and fill an important gap in our evaluation. We will incorporate the Direct QA ablation results and qualitative examples into the final version of the paper. We sincerely hope these results adequately address the remaining concerns.

---

### Decision · Program_Chairs · 2026-04-30

**Decision:**

Accept (spotlight)

**Comment:**

The authors tackles the spatial intelligence gap, and through some diagnostic experiments, they show the real bottleneck is not the visual encoder or the reasoning module, but the missing spatial interface in between. Their proposed solution teaches the model to internally generate three orthographic views as intermediate reasoning steps, and then reason over those self-generated mental sketches. The benchmark they built is constructed with both in-domain and out-of-domain splits. Overall the reviews were quite positive. The main concerns were around whether the gains really come from the framework itself or just from the new training data, how well the method generalizes beyond simple block-world settings, and whether the structured JSON output format is actually necessary. The authors handled the rebuttal well. The most convincing piece of new evidence was the direct QA ablation, where they trained the same base model on the exact same data but without any intermediate reasoning.